# ECHOPULSE: ECG CONTROLLED ECHOCARDIOGRAMS VIDEO GENERATION

**Yiwei Li**[1,2*]**, Sekeun Kim**[1*]**, Zihao Wu**[1,2†]**, Hanqi Jiang**[2†]**, Yi Pan**[2†]**,
Pengfei Jin**[1]**, Sifan Song**[1]**, Yucheng Shi**[1,2]**, Xiaowei Yu**[3]**, Tianze Yang**[2]**,
Tianming Liu**[1‡]**, Quanzheng Li**[2‡]**, Xiang Li**[2‡]
[1]Department of Radiology, Massachusetts General Hospital and Harvard Medical School
[2]School of Computing, University of Georgia
[3]Department of Computer Science and Engineering, University of Texas at Arlington

## ABSTRACT

Echocardiography (ECHO) is essential for cardiac assessments, but its video quality and interpretation heavily relies on manual expertise, leading to inconsistent results from clinical and portable devices. ECHO video generation offers a solution by improving automated monitoring through synthetic data and generating high-quality videos from routine health data. However, existing models often face high computational costs, slow inference, and rely on complex conditional prompts that require experts' annotations. To address these challenges, we propose ECHOPulse, an ECG-conditioned ECHO video generation model. ECHOPulse introduces two key advancements: (1) it accelerates ECHO video generation by leveraging VQ-VAE tokenization and masked visual token modeling for fast decoding, and (2) it conditions on readily accessible ECG signals, which are highly coherent with ECHO videos, bypassing complex conditional prompts. To the best of our knowledge, this is the first work to use time-series prompts like ECG signals for ECHO video generation. ECHOPulse not only enables controllable synthetic ECHO data generation but also provides updated cardiac function information for disease monitoring and prediction beyond ECG alone. Evaluations on three public and private datasets demonstrate state-of-the-art performance in ECHO video generation across both qualitative and quantitative measures. Additionally, ECHOPulse can be easily generalized to other modality generation tasks, such as cardiac MRI, fMRI, and 3D CT generation. Codes and examples can seen from `https://github.com/levyisthebest/ECHOPulse_Prelease`.

## 1 INTRODUCTION

Echocardiography (ECHO) is widely used for evaluating heart function and structure due to its real-time, non-invasive, and radiation-free nature (SF, 2009; Otto, 2013; Omar et al., 2016). Recent advancements in deep learning have shown promising performance in computer-aided cardiac diagnosis (Oktay et al., 2018; Smistad et al., 2020; Kim et al., 2021). However, echocardiography depends on experienced operators, leading to insufficient data for training robust models. Moreover, annotating ECHO data demands expert knowledge, making the process time-consuming and sometimes unavailable (Leclerc et al., 2019a; Lu et al., 2021).

One promising avenue is the use of video generation models to synthesize ECHO videos. In general domains, text-to-video (T2V) models (Yu et al., 2023; Zhang et al., 2024; Wang et al., 2024; Villegas et al., 2022) have been successful in creating realistic videos from text prompts. However, applying these models to the medical field remains challenging due to the necessity for precise control over anatomical structures, which cannot vary freely as in general domain. Previous controllable ECHO generation frameworks (Ashrafian et al., 2024; Zhou et al., 2024; Yu et al., 2024a) rely on carefully curated conditional prompts such as segmentation masks, clinical text, or optical flow, which only

---

[*]Co-first Authors.
[†]Co-Second Authors.
[‡]Corresponding author email: xli60@mgh.harvard.edu

experts can provide and fail to incorporate time-series information. Moreover, these diffusion-based models suffer from high computational costs and slow inference. Therefore, a fast and controllable ECHO generation framework without the need for expert prior conditions is highly desirable (Reynaud et al., 2024).

Inspired by these motivations, we propose ECHOPulse, an ECHO video generation model conditioned on electrocardiogram (ECG) signals. ECG data are readily accessible in both clinical and outpatient settings (Al-Khatib et al., 2018), and even from personal wearable devices (Liu et al., 2013). Since ECGs are typically the first step in evaluating cardiac issues before performing further echocardiography (Al-Khatib et al., 2018; Del Torto et al., 2022), they serve as a natural and intrinsic prior for coherent ECHO video generation. By leveraging ECG signals, we can generate ECHO videos without the need for complex conditional prompts or expert annotations. To the best of our knowledge, this is the first work to utilize time-series prompt for ECHO video generation. In clinical scenarios where prior ECHO images exist, ECHOPulse can be co-conditioned on both ECG and prior imaging to provide updated cardiac function information, such as left ventricular ejection fraction (LVEF/EF) for cardiac disease monitoring and prediction (Ponikowski et al., 2016), enhancing computer-aided diagnosis beyond ECG alone.

Our main contributions are:

- We propose ECHOPulse, a fast and controllable ECHO video generation framework conditioned on ECG for the first time, which also shows potential to generalize to other modality generation tasks, such as cardiac MRI, fMRI, and 3D CT generation.

- By utilizing readily available ECG data for controllable and precise ECHO generation personalized for each patient, we bypass the need for paired expert annotations, showing potential for scaling to larger video datasets.

- Evaluations on two public and one private dataset demonstrate that our framework achieves SOTA performance in both quantitative and qualitative metrics.

- We collect a synthetic ECHO video dataset consisting of ECHO videos paired with ECG signals and segmentation masks, which we plan to release to enable researchers to evaluate the practicality and reliability of synthetic ECHO video generation.

## 2 RELATED WORK

### 2.1 VISUAL TOKENIZATION

Visual Tokenization is crucial for converting images or videos into discrete tokens that can be processed by language models. One of the foundational approaches to visual tokenization is VQ-VAE (Van Den Oord & Vinyals, 2017), which uses a convolutional neural network (CNN) to encode visual data, followed by a vector quantization (VQ) step. This method assigns each embedding to the closest entry in a codebook, converting continuous embeddings into discrete tokens. Video tokenization presents greater challenges, and VQGAN has been adapted to address these difficulties (Villegas et al., 2022; Yu et al., 2023). The current state-of-the-art in video tokenization is MAGVIT2 (Yu et al., 2024b), which employs an enhanced 3D architecture, leverages an inflation technique for initialization based on image pre-training, and incorporates robust training losses to improve performance.

### 2.2 VIDEO GENERATION

Recent video generation advances have increasingly focused on discrete representations to manage the high dimensionality of video data. Models like VQ-VAE (Van Den Oord & Vinyals, 2017) convert continuous video frames into discrete vector indices, reducing data complexity and enabling more efficient sequence modeling. Phenaki (Villegas et al., 2022) uses a causal Transformer with discrete temporal embeddings for variable-length video generation, while Masked Generative Transformer (Chang et al., 2022) leverages discrete techniques for sparse spatiotemporal relationships. These studies highlight the advantages of discrete representations in balancing complexity and efficiency, especially for long sequences. Moreover, autoregressive models with discrete tokens (Yu et al., 2022) generate diverse, high-quality content, and discrete diffusion models (Austin et al.,

2021) extend the diffusion process to the discrete symbol space, enhancing scalability and long-range video modeling.

## 2.3 ECHO VIDEO GENERATION

Ultrasound imaging is a non-invasive, real-time, and portable medical imaging modality that plays a crucial role in clinical diagnostics. However, the quality and interpretation of ultrasound images heavily depend on the operator's expertise, and obtaining high-quality ultrasound datasets is challenging. Consequently, the generation of ultrasound images has attracted considerable attention in recent years. Recent works, such as HeartBeat (Zhou et al., 2024), have leveraged deep learning techniques like diffusion models (Ho et al., 2020) to achieve controllable synthesis of echocardiograms. By incorporating multimodal conditions—such as cardiac anatomical annotations and functional indices—and employing diffusion models, these methods can generate realistic ultrasound videos with specific cardiac morphology and function. This approach enriches training datasets and enhances algorithm robustness. Ouyang et al. (2020) introduced the EchoNet-Dynamic model, which utilizes spatiotemporal three-dimensional convolutional neural networks to segment the left ventricle frame by frame and estimate the ejection fraction (EF), outperforming manual measurements. Such advancements hold promise for assisting clinicians in more precise and efficient cardiac function assessments. These techniques typically use images as conditional inputs to generate corresponding ultrasound images. Recent studies have also explored generating ultrasound video sequences from single-frame images (Reynaud et al., 2023), offering new possibilities for dynamic ultrasound applications like cardiac function evaluation. However, these studies rely on carefully curated conditional prompts, such as segmentation masks and clinical text (Zhou et al., 2024), or require complex procedures and expert annotations for data acquisition, as seen with Ejection Fraction (EF) (Reynaud et al., 2023; 2024), yet they do not leverage time-series ECG data. In contrast, our work is the first to efficiently generate controllable ECHO videos conditioned on ECG data.

## 3 METHODS

ECHOPulse, as shown in Figure 1, consists of three key components: (1) a video tokenization model that encodes videos into temporal token sequences, (2) a masked generative transformer that learns the latent alignment between video tokens and ECG signal, and (3) a progressive video generation pipeline that enables fast, unlimited video generation. The video tokenization and transformer components are trained separately.

### 3.1 VIDEO TOKENIZATION

The key to generating videos from ECG signals lies in capturing the temporal correlation between the video and ECG data. Unlike text (Touvron et al., 2023), ECG is a continuous temporal signal with variable sequence lengths. As shown in Figure 2, each ECG phase should correspond to different stages of the ECHO. Thus, encoding the entire video into a single embedding risks disrupting temporal alignment between ECG and video, also complicating autoregressive video generation and reducing generalizability. To temporally align ECHO video with the various phases of the ECG and support autoregressive video generation, an effective approach is to discretize the video into tokens while splitting the ECG into patches and embedding them for cross-modal alignment. Drawing inspiration from ViViT (Arnab et al., 2021) and C-ViViT (Villegas et al., 2022), we developed ECHOPulse's video tokenization model to meet these specific requirements.

**Model architecture:**   The encoding starts with a video sequence of $T + 1$ frames, each with resolution $H \times W$ and $C$ channels, denoted as $\mathbf{X} \in \mathbb{R}^{(T+1) \times H \times W \times C}$. The sequence is compressed into tokens of size $(T' + 1) \times H' \times W'$, where the first frame is tokenized independently, and subsequent frames are spatio-temporal tokens that autoregressive depending on the previous. Non-overlapping patches of size $P_h \times P_w \times C$ (for images) and $P_t \times P_h \times P_w \times C$ (for video) are extracted, resulting in $T' = \frac{T}{P_t}$, $H' = \frac{H}{P_h}$, and $W' = \frac{W}{P_w}$. Each patch is flattened and projected into a $D$-dimensional space, forming a tensor of shape $(T' + 1) \times (H' \times W') \times D$. Spatial transformers are applied with self attention, followed by temporal transformers with causal attention. The patch embeddings $\mathbf{Z} \in \mathbb{R}^{(T'+1) \times H' \times W' \times D}$ are then quantized into codewords $\mathbf{C_z}$ via vector quantization. The decoder

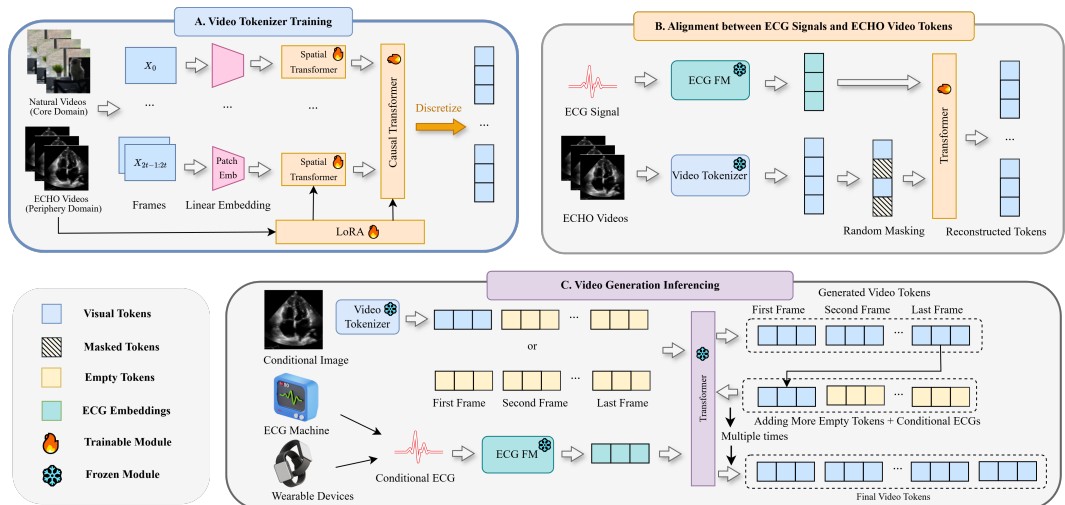

Figure 1: The pipeline of the ECHOPulse. ECHOPulse contains a two-step training procedure. a) The first step trains the video tokenizer on the natural video dataset first and fine-tunes it on the public ECHO video dataset. b) The second step trains the transformer via the input token produced by ECG foundation model and video tokenizer, pretrained in the first step. Followed by the video generation procedure (c), ECHOPulse accepts input with or without a conditional image. The empty tokens will be reconstructed via the frozen transformer, trained in the second step, through the guidance of ECG siganls. ECHOPulse is capable of generating continuous long videos by sequentially shifting the token sequence and integrating new ECG inputs.

reverses the process by transforming quantized tokens into embeddings, applying temporal and spatial transformers, and mapping the tokens back to pixel space via linear projection to reconstruct the frames $\hat{\mathbf{X}}$.

Additionally, Our framework incorporates the Lookup-Free Quantization (LFQ) method, inspired by the advancements in MAGVIT2 (Yu et al., 2024b), to streamline the quantization process. Unlike traditional vector quantization that relies on an explicit codebook lookup, LFQ parameterizes the quantization operation directly. Moreover, due to the lack of high-quality medical video datasets comparable in scale to natural video datasets, we introduce LoRA (Low-Rank Adaptation) (Hu et al., 2021) to enhance the learning efficiency and generalizability for medical video generation. This also lays the foundation for offering personalized video generation services for users or patients in future applications.

**Loss design:** During video tokenization training phrase, the total loss function $\mathcal{L}_{\text{total}}$ is formulated as:

$$\mathcal{L}_{\text{total}} = \mathcal{L}_{\text{recon}} + \mathcal{L}_{\text{percep}} + \mathcal{L}_{\text{VQ}} + \lambda_{\text{adaptive}} \cdot \mathcal{L}_{\text{GAN}}, \tag{1}$$

where $\mathcal{L}_{\text{recon}}$ is the reconstruction loss, $\mathcal{L}_{\text{percep}}$ is the perceptual loss, $\mathcal{L}_{\text{VQ}}$ represents the vector quantization loss, and $\mathcal{L}_{\text{GAN}}$ Gu et al. (2022) is the generator adversarial loss. The parameter $\lambda_{\text{adaptive}}$ dynamically adjusts the contribution of the adversarial loss relative to the perceptual loss during training.

Specifically, the reconstruction loss $\mathcal{L}_{\text{recon}}$ is computed using the Mean Squared Error (MSE) to ensure the reconstructed video $\hat{\mathbf{V}}$ to closely match the original video $\mathbf{V}$ at the pixel level:

$$\mathcal{L}_{\text{recon}} = \frac{1}{N} \left\| \mathbf{V} - \hat{\mathbf{V}} \right\|_2^2, \tag{2}$$

where $N$ is the total number of pixels across all frames, $\mathbf{V}_i$ and $\hat{\mathbf{V}}_i$ are the $i$-th pixel of the original video and the reconstructed video, respectively.

To improve the perceptual quality of the reconstructed video, perceptual loss $\mathcal{L}_{\text{percep}}$ is employed to measure the similarity (Johnson et al., 2016) between the original and reconstructed videos in the

feature space of a pre-trained VGG16 (Qassim et al., 2018) network $\phi$. By extracting high-level features from randomly selected video frames $\mathbf{V}_{f_j}$ and $\hat{\mathbf{V}}_{f_j}$, the perceptual loss is defined as:

$$\mathcal{L}_{\text{percep}} = \frac{1}{M} \sum_{j=1}^{M} \left\| \phi(\mathbf{V}_{f_j}) - \phi(\hat{\mathbf{V}}_{f_j}) \right\|_2^2, \tag{3}$$

where $M$ is the number of feature elements, and $\phi(\cdot)$ denotes the VGG feature extraction operation.

The vector quantization loss (van den Oord et al., 2017) $\mathcal{L}_{\text{VQ}}$ regularizes the encoder outputs by ensuring that they are close to their nearest codebook entries, encouraging efficient and discrete representation learning. The VQ loss is defined as:

$$\mathcal{L}_{\text{VQ}} = \beta \cdot \left\| \text{sg}[\mathbf{z}_e] - \mathbf{e} \right\|_2^2, \tag{4}$$

where $\mathbf{z}_e$ is the encoder output, $\mathbf{e}$ is the closest codebook entry, and $\beta$ is a weighting parameter. The term $\text{sg}[\cdot]$ represents the stop-gradient operation, which ensures that the encoder is encouraged to commit to a discrete codebook entry without backpropagating gradients through this operation.

The generator adversarial loss $\mathcal{L}_{\text{GAN}}$ drives the model to produce videos that are indistinguishable from real videos, as evaluated by a discriminator $D$. We employ the hinge loss formulation for stable adversarial training:

$$\mathcal{L}_{\text{GAN}} = -\mathbb{E}_{\hat{\mathbf{V}} \sim P_G} \left[ D(\hat{\mathbf{H}}) \right], \tag{5}$$

where $\hat{\mathbf{H}}$ is a video generated by the model, sampled from the generator's distribution $P_G$, and $D(\hat{\mathbf{H}})$ is the discriminator's output for the generated video.

To balance the adversarial loss $\mathcal{L}_{\text{GAN}}$ and the perceptual loss $\mathcal{L}_{\text{percep}}$, we introduce an adaptive weight $\lambda_{\text{adaptive}}$. This weight is computed dynamically based on the gradients of both losses with respect to the parameters of the decoder's last layer (Johnson et al., 2016). The adaptive weight is calculated as:

$$\lambda_{\text{adaptive}} = \frac{\|\nabla_\theta \mathcal{L}_{\text{percep}}\|_2}{\|\nabla_\theta \mathcal{L}_{\text{GAN}}\|_2 + \epsilon}, \tag{6}$$

where $\theta$ are the parameters of the decoder's last layer, $\nabla_\theta \mathcal{L}_{\text{percep}}$ and $\nabla_\theta \mathcal{L}_{\text{GAN}}$ represent the gradients of the perceptual and adversarial losses with respect to $\theta$, and $\epsilon$ is a small constant to avoid division by zero.

## 3.2 Alignment between Video and ECG Tokens

Next, we embed the ECG signals and align them with the video tokens. To fully leverage the time-series ECG data, we implemented the following encoding and alignment design.

**ECG encoder:** To better couple with echo video, the time-series ECG is first divided into non-overlapping temporal and spatial patches, which are then linearly projected into a high-dimensional space (Na et al., 2024). Self-attention mechanisms capture spatio-temporal relationships across time segments, and a masked autoencoder (MAE) (He et al., 2022) further refines the model's understanding of ECG dynamics, transforming them into separate embedding features. In this work, we divide ECG signals into patches using the embedding layer of an ECG foundation model (ECG-FM), whose strong predictive performance ensures the effectiveness of this patch-based representation. The proposed patchify approach is as follows: The raw ECG signal, denoted as $\mathbf{X} \in \mathbb{R}^{L \times T}$, where $L$ is the number of leads and $T$ is the time length, is split into non-overlapping spatio-temporal patches. Each lead $l$ produces patches $\mathbf{P}^l \in \mathbb{R}^{n \times p}$, where $n = \frac{T}{p}$ and $p$ is the patch size.

Each patch $\mathbf{P}_i^l$ is passed through the ECG-FM encoder, producing a $D$-dimensional embedding:

$$\mathbf{Z}_i^l = \text{ECG-FM}(\mathbf{P}_i^l) \in \mathbb{R}^D,$$

which results in a sequence of embeddings $\mathbf{Z} = \{\mathbf{Z}_1^l, \mathbf{Z}_2^l, \dots\} \in \mathbb{R}^{L \times n \times D}$.

**Aligning ECG and video token sequences for video prediction:** Our video generation framework employs a bidirectional transformer with MVTM (Chang et al., 2022) to predict video tokens. We mask tokens during training and use parallel decoding during inference to iteratively fill in missing tokens based on context from all directions. This design allows for faster and more efficient video synthesis, leveraging the bidirectional self-attention mechanism to condition video generation on both past and future frames. The masked token prediction is driven by minimizing the negative log-likelihood of the masked tokens:

$$\mathcal{L}_{\text{mask}} = -\mathbb{E}_{\mathbf{X}} \left[ \sum_{i=1}^{N} m_i \log P(\mathbf{y}_i | \mathbf{Y}_M) \right], \tag{7}$$

where $\mathbf{y}_i$ are the masked tokens, $m_i$ indicates the mask for each token, and $\mathbf{Y}_M$ represents the partially masked token sequence.

At each iteration, the mask scheduling function $\gamma(t/T)$ determines the fraction of tokens to mask:

$$\gamma(t/T) = \cos\left(\frac{\pi t}{2T}\right), \tag{8}$$

where $t$ is the current iteration and $T$ is the total number of iterations. The cosine schedule progressively reduces the mask ratio, enabling a gradual refinement of the generated video (Chang et al., 2022).

The bidirectional transformer architecture allows for flexible video generation guided by various conditioning inputs. We also adapted T5X (Roberts et al., 2023) following (Reynaud et al., 2024; Ashrafian et al., 2024)'s text-based ECHO video generation method to create a text-to-ECHO video version for comparative experiments. What's more, we introduce an ECG mask similar to a text mask. This mask ensures that certain portions of the ECG signal are masked out during training, forcing the model to learn to generate corresponding video frames based on the visible ECG features.

**Optimization:** To ensure the transformer model fully learns the entire action of ECHO videos, we add a critic loss function (Lezama et al., 2022). The critic takes a sequence of predicted tokens $\hat{\mathbf{y}}$ and real tokens $\mathbf{y}$ and computes whether the predicted tokens are real or fake. The critic's objective is expressed as:

$$\mathcal{L}_{\text{critic}} = -\mathbb{E}_{\mathbf{y}} \left[ \log D(\mathbf{y}) \right] - \mathbb{E}_{\hat{\mathbf{y}}} \left[ \log(1 - D(\hat{\mathbf{y}})) \right], \tag{9}$$

where $D(\mathbf{y})$ is the critic's prediction for the real tokens, and $D(\hat{\mathbf{y}})$ is its prediction for the fake tokens (predicted by transformer). In our framework, $\mathbf{y}$ represents the actual video tokens, and $\hat{\mathbf{y}}$ are the predicted tokens sampled during the transformer reconstruction process.

In practice, the critic's loss function is computed using the following binary cross-entropy (BCE) loss (Goodfellow, 2016):

$$\mathcal{L}_{\text{BCE}} = -\frac{1}{N} \sum_{i=1}^{N} \left[ y_i \log(\hat{y}_i) + (1 - y_i) \log(1 - \hat{y}_i) \right], \tag{10}$$

where $y_i$ are the true labels (1 for real tokens, 0 for fake tokens) and $\hat{y}_i$ are the critic's predicted probabilities.

### 3.3 VIDEO GENERATION

We first train a video tokenization model and an ECG-conditioned video token prediction transformer. To generate videos conditioned on ECG, we pass the empty video value through the video tokenization model to obtain empty tokens and use the ECG tokens as input to the pre-trained transformer. This mirrors the training setup where all video tokens are masked, enabling video prediction from ECG. Additionally, as the first frame is independently tokenized, using it as a condition mimics the training scenario where only the first frame is visible, allowing for video generation based on the initial frame. Similarly, by conditioning on the final segment of a previous video and another ECG inputs, the transformer can generate coherent subsequent frames.

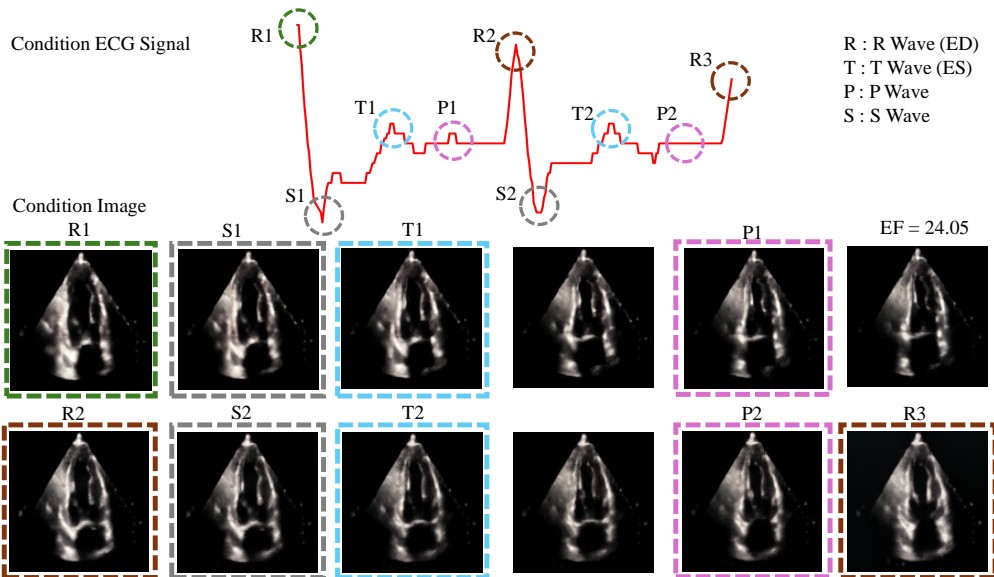

Figure 2: Video generation example of ECHOPulse. The inputs to ECHOPulse consist of a conditioning image and an ECG signal. Utilizing these inputs as constraints, ECHOPulse generates corresponding videos. The RSTP waves represent the four phases of the ECG. The R wave corresponds to end-diastole (ED), which is the frame with the largest ventricular segmentation area, while the T wave corresponds to end-systole (ES), the frame with the smallest ventricular area. In this case the EF of the generated ECHO video is 24.05.

**Progressive video generation with auto-regressive extrapolation:** During inference, video tokens are sampled iteratively, similar to the process described in MaskGIT (Chang et al., 2022), using classifier-free guidance with a scaling factor $\lambda$ to balance the alignment between the generated video and the provided ECG condition. Once the first $t_x + 1$ frames are generated in the latent space, we can further extrapolate additional frames in an auto-regressive manner by re-encoding the last $K$ generated frames using the video encoder. These tokens are then used to initialize the pre-trained transformer, which generates the remaining video tokens conditioned on the provided ECG. This approach enables rapid and length-unconstrained generation of ECHO videos, significantly enhancing its clinical applicability.

## 4 EXPERIMENTS

**Dataset and implementations:** In this experiment, the process is conducted in a distributed manner. Initially, we train the video tokenization model on the Webvim-10M dataset (Bain et al., 2021), followed by fine-tuning on the CAMUS dataset (Leclerc et al., 2019b) to adapt the tokenization for medical imaging tasks. With the introduction of LFQ, the codebook size is set to $2^{13}$. All natural videos are resized to 128x128 resolution with 11 frames. The temporal patch size is set to 2, and the spatial patch size is set to 8, resulting in a token sequence with a length of 1536. More detailed information can be seen in Appendix Table 4 and Table 5.

For the subsequent training of ECHOPulse, we utilized a private dataset consisting of 94,078 video and ECG samples. A total of 473 videos were extracted from the dataset to serve as the test dataset, and all video generation evaluations were performed on this test dataset. During the training process, all ECHO videos are resized to a format of 128×128 with 11 frames in the dataloader to match the input requirements of the video encoder. Due to the large volume and diverse sources of our dataset, the overall quality is not as consistent as that of publicly available ECHO datasets. Therefore, each video undergoes random contrast augmentation during loading to enhance its visual quality. We used the pretrained weight of ST-MEM (Na et al., 2024) as ECG signal encoder. Similarly, the evaluation metrics were maintained to ensure a fair and direct comparison. This allowed us to evaluate the model's performance and validate the effectiveness of the proposed method.

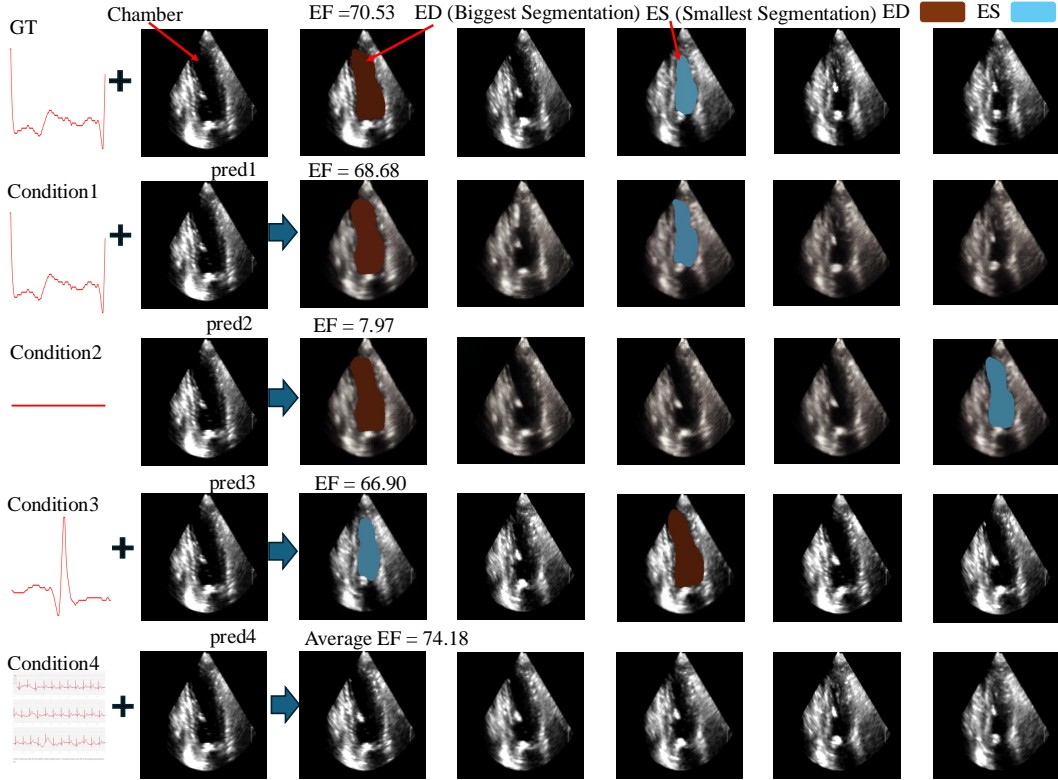

Figure 3: Video generation results from ECHOPulse under different ECG conditions. The first row displays the ground truth, while the second row shows the generated video using the same ECG. The third row illustrates results under flat ECG conditions. The fourth condition depicts a shifted R wave, and the final condition, collected from an Apple Watch, represents the most common scenario. The ECG for Condition 4 was collected directly from the Health app on Apple Watch v9.

For evaluation, we use the CAMUS (Leclerc et al., 2019a) and EchoNet-Dynamic (Ouyang et al., 2020) datasets. CAMUS, a public dataset with 500 ECHO videos with diverse spatial resolutions, is currently the highest-quality public ECHO dataset available. EchoNet-Dynamic includes 10,030 4-chamber cardiac ultrasound sequences with a resolution of 112×112 pixels.

All experiments were conducted on 8 NVIDIA H-100 GPUs, each with 80GB of memory. Due to potential privacy concerns, the private dataset used in this study cannot be made publicly available. However, we have released all the code, scripts, and the pretrained ECG2Video model weights to facilitate further research and replication of our work.

## 4.1 VIDEO TOKENIZATION RECONSTRUCTION PERFORMANCE

In our experiments, we conduct a quantitative comparison between the performance of the diffusion-based EchoNet-Synthetic model and our VQ-VAE-based model, primarily evaluated using MSE (Mean Squared Error) (Gauss, 1809) and MAE (Mean Absolute Error) (Qi et al., 2020) on reconstruction results of different datasets. As shown in Table 1.

Specifically, the EchoNet-Synthetic model achieves an MSE of $3.00 \times 10^{-3}$ and an MAE of $3.00 \times 10^{-2}$ on the Echo-Dynamic dataset. In comparison, our VQ-VAE based model, initially trained on a natural video dataset, exhibits an MSE of $5.02 \times 10^{-3}$ and an MAE of $4.08 \times 10^{-2}$ on the same dataset. Although the initial errors are slightly higher, after only one epoch of fine-tuning, the MSE of our model significantly improves to $2.89 \times 10^{-3}$, with the MAE reduced to $2.92 \times 10^{-2}$, surpassing the performance of the diffusion-based model.

Table 1: **Reconstruction metrics:** Videos reconstructed using video tokenization and diffusion-based models are evaluated across two datasets with MAE and MSE. The best results are highlighted.

| Methods | Dataset | MSE↓ | MAE↓ |
|---|---|---|---|
| EchoNet-Synthetic (Reynaud et al., 2024) | Echo-Dynamic | $3.00 \times 10^{-3}$ | $3.00 \times 10^{-2}$ |
| ECHOPulse (Only natural videos) | Echo-Dynamic | $5.02 \times 10^{-3}$ | $4.08 \times 10^{-2}$ |
| ECHOPulse (Domain transfer) | Echo-Dynamic | $2.89 \times 10^{-3}$ | $2.92 \times 10^{-2}$ |
| EchoNet-Synthetic (Reynaud et al., 2024) | Echo-Dynamic | $3.12 \times 10^{-3}$ | $3.37 \times 10^{-2}$ |
| ECHOPulse (Only natural videos) | Private data | $6.09 \times 10^{-3}$ | $4.41 \times 10^{-2}$ |
| ECHOPulse (Domain transfer) | Private data | $2.91 \times 10^{-3}$ | $2.97 \times 10^{-2}$ |

Table 2: **Image quality metrics:** The videos generated by different methods under various conditions across three datasets are evaluated using three image quality metrics. The best results are highlighted. Note that EchoNet-Synthetic (Reynaud et al., 2024) did not report the SSIM metric and did not specify whether the results correspond to A2C or A4C views.

| Methods | Dataset | Condition | A2C | | | A4C | | |
|---|---|---|---|---|---|---|---|---|
| | | | FID↓ | FVD↓ | SSIM↑ | FID↓ | FVD↓ | SSIM↑ |
| MoonShot (Zhang et al., 2024) | CAMUS | Text | 48.44 | 202.41 | 0.63 | 61.57 | 290.08 | 0.62 |
| VideoComposer (Wang et al., 2024) | CAMUS | Text | 37.68 | 164.96 | 0.60 | 35.04 | 180.32 | 0.61 |
| HeartBeat (Zhou et al., 2024) | CAMUS | Text | 107.66 | 305.12 | 0.53 | 76.46 | 381.28 | 0.53 |
| HeartBeat (Zhou et al., 2024) | CAMUS | Text+Sketch+Mask+... | 25.23 | 97.28 | 0.66 | 31.99 | 159.36 | 0.65 |
| ECHOPulse (Only natural videos) | CAMUS | Text(EF) | 12.71 | 273.15 | 0.61 | 15.38 | 336.04 | 0.58 |
| ECHOPulse (Domain transfer) | CAMUS | Text(EF) | 5.65 | 211.85 | 0.79 | 8.17 | 283.32 | 0.75 |
| EchoDiffusion (Reynaud et al., 2023) | Echo-Dynamic | EF | - | - | - | 24.00 | 228.00 | 0.48 |
| EchoNet-Synthetic (Reynaud et al., 2024) | Echo-Dynamic | EF | - | - | - | 16.90 | 87.40 | - |
| ECHOPulse (Only natural videos) | Echo-Dynamic | Text(EF) | - | - | - | 40.18 | 317.03 | 0.37 |
| ECHOPulse (Domain transfer) | Echo-Dynamic | Text(EF) | - | - | - | 28.25 | 296.60 | 0.43 |
| EchoDiffusion (Reynaud et al., 2023) | Private data | EF | 20.71 | 379.43 | 0.55 | 23.20 | 390.17 | 0.53 |
| EchoNet-Synthetic (Reynaud et al., 2024) | Private data | EF | 18.39 | 91.29 | 0.56 | 26.13 | 120.91 | 0.55 |
| ECHOPulse (Only natural videos) | Private data | Text(EF) | 27.49 | 291.67 | 0.53 | 34.13 | 374.92 | 0.51 |
| ECHOPulse(Domain transfer) | Private data | Text(EF) | 25.44 | 224.90 | 0.54 | 31.21 | 334.09 | 0.54 |
| ECHOPulse (Only natural videos) | Private data | ECG | 18.73 | 200.45 | 0.56 | 27.37 | 302.89 | 0.55 |
| ECHOPulse (Domain transfer) | Private data | ECG | 15.50 | 82.44 | 0.67 | 20.82 | 107.40 | 0.66 |

On the private dataset, the results followed a similar trend. Before fine-tuning, the VQ-VAE model had an MSE of $6.09 \times 10^{-3}$ and an MAE of $4.41 \times 10^{-2}$. However, after one epoch of domain transfer, the MSE was significantly reduced to $2.91 \times 10^{-3}$, and the MAE to $2.97 \times 10^{-2}$.

These results indicate that, despite the common expectation that diffusion-based models perform better in terms of reconstruction quality, our VQ-VAE model was able to quickly converge in a new domain. With only one epoch of fine-tuning, it achieved results comparable to or even surpassing those of the diffusion-based model's VAE in this task. This further demonstrates the potential of our model for clinical applications, where it can quickly adapt to new tasks or datasets without the need for retraining from scratch, offering greater practical utility and flexibility.

## 4.2 VIDEO GENERATION PERFORMANCE

We evaluate our model on two objectives: 1) image quality (FID (Heusel et al., 2017), FVD (Unterthiner et al., 2018), SSIM (Wang et al., 2004)) and left ventricular ejection fraction (LVEF/EF) accuracy. Since we are the first to explore ECG-guided ECHO video generation, and some prior works generating ECHO videos from text have not been open-sourced, we were unable to reproduce their results on our dataset. However, those related papers have clearly reported experimental parameters, datasets, and data split methods. Therefore, we trained ECHOPulse using T5X (Roberts et al., 2023) as the text encoder on both the CAMUS and EchoNet datasets. All experimental settings, including video resolution and parameters, were kept consistent with those reported in previous studies. Details can be seen in Appendix A.2.3.

Based on Table 2, ECHOPulse outperforms other models across several key metrics. In the ECG-conditioned setting, it achieves the lowest FID scores of 15.50 (A2C) and 20.82 (A4C), indicating highly realistic video generation. Its FVD scores of 82.44 (A2C) and 107.40 (A4C) further highlight its ability to capture temporal dynamics. Additionally, ECHOPulse's SSIM scores of 0.67 (A2C) and 0.66 (A4C) are higher than those of competing models, showcasing its superior video quality and structural coherence.

Table 3: **Clinical and time-inference metrics:** This table presents the EF (Ejection Fraction) (Murphy et al., 2020) differences between the generated videos and the target ones. The sampling time refers to the time required to generate 64 video frames. The best results are highlighted.

| Methods | Condition | $R^2 \uparrow$ | MAE $\downarrow$ | RMSE $\downarrow$ | S. time$\downarrow$ | Parameters |
|---|---|---|---|---|---|---|
| EchoDiffusion (Reynaud et al., 2023) | Image+EF | 0.83 | 5.14 | 7.21 | Gen. 147s | 402M |
| EchoNet-Synthetic (Reynaud et al., 2024) | Image+EF | 0.85 | 2.71 | 3.13 | Gen. 20.2s | 283M |
| ECHOPulse (Our Model) | Image+ECG | 0.85 | 2.51 | 2.86 | Gen. 6.4s | 279M |

ECHOPulse+Text significantly outperforms the HeartBeat model, which also uses text control, on the CAMUS dataset, achieving lower FID 5.65/8.17 and higher SSIM scores 0.79/0.75. As CAMUS is currently the highest-quality publicly available ECHO dataset, all models perform better on CAMUS, while the modest FVD scores 211.85/283.32 highlight the limitations of text-based control. ECHOPulse+Txt also exceeds EchoDiffusion on EchoNet-Dynamic. While VQ-VAE models may lag behind diffusion in video quality, their compatibility with ECG foundation models, coupled with the strong ECG-ECHO correlation, drives ECHOPulse's superior performance.

After LoRA fine-tuning, ECHOPulse shows further improvement but still falls slightly short of HeartBeat, which uses six conditions, and EchoNet-Synthetic, which generates entire videos based on a given LVEF value and condition image. However, ECHOPulse+ECG excels on the larger, more complex private dataset, demonstrating the effectiveness of ECG as a condition for video generation. Comparison examples can be found in Appendix Figure 8 and 9.

To validate that the generated videos are synchronized with the ECG signals, we compared the cardiac phases between the generated and original videos, focusing on the end-diastole (ED) and end-systole (ES) phases. ED, which corresponds to the R wave in the ECG, represents the moment when the chamber area is at its largest. Similarly, ES corresponds to the T wave, when the chamber area is at its smallest. As shown in Figure 2, the generated videos, conditioned on the input image and ECG, strictly follow the phase changes dictated by the ECG. A more quantitative evaluation is performed by assessing the LVEF, demonstrating the alignment of cardiac dynamics between the generated and original ECHO video.

To evaluate our model's performance on LVEF accuracy, we applied SAM2 (Ravi et al., 2024) to the generated videos for direct segmentation of the endocardium, epicardium, and left atrium. After segmenting the cardiac chamber structures, we compared the LVEF between the original echocardiograms and the generated ones. $LVEF = \frac{LVED - LVES}{LVED}$. This comparison ensured that the generated videos were controlled solely by the ECG signals—introducing temporal information without affecting the structural integrity of the heart images. As shown in Figure 3. Relevant information can be found in the Appendix A.3. Expert evaluation results and more examples can be seen in Appendix A.4

It can be observed that ECHOPulse is capable of generating realistic videos with relatively accurate EF values, regardless of whether the input is an original ECG or a flat ECG. In particular, Condition 4 demonstrates the model's ability to generate videos using daily ECG signals collected directly from an Apple Watch. This highlights ECHOPulse's zero-shot capability in handling general ECG signals, showcasing its potential for broad applicability with real-world ECG data.

Quantitative results of $LVEF$ are provided in Table 3, where our model outperforms others (Reynaud et al., 2023; 2024) in terms of R square MAE and RMSE. ECHOPulse achieves these results using less sampling time to generate higher-quality videos. This not only underscores the validity of using ECG as a conditioning signal but also demonstrates the clinical potential of our model.

## 5 CONCLUSION

In this paper, we present ECHOPulse, an innovative framework for generating ECHO videos conditioned on ECG signals, marking a significant advancement in the integration of time-series data for video synthesis. By utilizing VQVAE-based tokenization and a transformer with MVTM-driven generation, ECHOPulse not only achieves superior performance in video quality but also addresses key challenge of ejection fraction estimation in cardiac video generation. Our extensive evaluation on both public and private datasets highlights the model's robust potential for enhancing clinical

workflows and enabling real-time cardiac monitoring. This work lays the foundation for further exploration in automated, data-driven cardiac assessment, with implications for both research and practical applications.

## 6    LIMITATION AND FUTURE WORK

There are several limitations of this study that have yet to be addressed. Although we have expanded the dataset through the generation of synthetic data, we are still unable to automatically ensure that the generated videos are free from potential personal privacy information, which limits our ability to fully release the whole private dataset. Additionally, while ECHOPulse can generate videos of unlimited length, it currently lacks the ability to improve video frame rates (FPS) and resolution. In the future, integrating. ECHOPulse, when combined with diffusion models and techniques like super-resolution and denoising, could further enhance the quality of generated videos. This approach addresses current limitations and pushes the boundaries of ECHO video generation.

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

# A APPENDIX

## A.1 REPRODUCIBILITY STATEMENT

To ensure the reproducibility of our results, we will make all relevant resources publicly available. This includes the complete codebase, training scripts, pre-trained model weights, and synthesized ECHO video data. By providing these materials, we encourage the research community to replicate and build upon our work. Detailed instructions on how to set up and run the experiments will also be included in our repository to facilitate easy reproduction of our results.

## A.2 MODEL SETUP AND HYPERPARAMETERS

### A.2.1 VIDEO TOKENIZER

The configuration of the video tokenizer model is as follows:

| Parameter | Value |
|---|---|
| Video input | 11 frames, frame stride 1, $128 \times 128$ resolution |
| Batch size | 128 |
| Embedding size | 512 |
| Codebook size | 8192 |
| Spatial patch size | 8 (non-overlapping) |
| Temporal patch size | 2 (along the temporal axis) |
| Spatial transformer depth | 4 layers |
| Temporal transformer depth | 4 layers |
| Transformer head dimension | 64 |
| Number of heads | 8 |
| Feed-forward multiplier | 4 (MLP size of 2048, i.e., $32 \times 64$) |

Table 4: Video tokenizer hyperparameters

### A.2.2 VIDEO GENERATION TRANSFORMER

The configuration of the transformer model in Figure 1b is as follows:

| Parameter | Value |
|---|---|
| Number of tokens | 8192 (aligned with the codebook size) |
| Batch size | 128 |
| Embedding dimension | 512 |
| ECG signal dimension | 768 |
| Depth (number of layers) | 6 |
| Learning rate | 1e-4 |
| Training steps | 4,000,000 steps |
| Exponential Moving Average (EMA) | Updated every 10 steps with a decay rate of 0.995 |
| Optimizer | Adam optimizer with $\beta$ values set to (0.9, 0.99) |

Table 5: Video generation model hyperparameters

### A.2.3 EXPERIMENTAL DETAILS

In the comparative experiments with Heartbeat, we encountered challenges in directly reproducing its results due to the lack of publicly available source code and detailed parameter specifications. Similarly, other generic methods compared in the Heartbeat paper, such as MoonShot and Video-Composer, were fine-tuned or retrained on the CAMUS dataset, but the paper does not provide sufficient implementation details to facilitate replication. However, Heartbeat explicitly specifies key aspects such as the video resolution, certain parameters, and the use of the CAMUS dataset with

Table 6: **Video reconstruction metrics:** This table presents the (reconstruction FID) rFID and (reconstruction LVEF) rLVEF differences between the reconstructed videos and the target ones.

| Methods | rFID↓ | rLVEF |
|---|---|---|
| EchoNet-Synthetic(Reynaud et al., 2024) | 10.93 | $1 \pm 0.33$ |
| ECHOPulse (Our Model) | 9.16 | $1 \pm 0.26$ |

its defined train, test, and validation splits. To ensure a fair comparison, we aligned the experimental conditions by training EchoPulse under the same settings as Heartbeat. Additionally, the text conditions for EchoPulse were configured to match those described in the Heartbeat study.

For the comparative experiments with EchoDiffusion and EchoNet-Synthetic on 8 NVIDIA H100s, we first trained EchoPulse on the EchoNet-Dynamic dataset, adhering to the same video format used in these methods. Specifically, the videos were resized to the 112×112 resolution standard of the EchoNet-Dynamic dataset. To ensure consistency with these approaches, we adapted our text condition to represent EF values in textual format, allowing EF data to serve as input to the model. However, converting numerical values into textual representations may have introduced some impact on the training effectiveness.

For experiments on the private dataset, we replicated the results of EchoDiffusion and EchoNet-Synthetic by following the training procedures described in their respective open-source codebases and publications. Subsequently, we compared the models' performance using the test set of the private dataset.

For the computation of metrics such as FVD, FID, and SSIM, we followed the approach outlined in EchoNet-Synthetic to enable comparison between generated videos and Ground-Truth videos. Specifically, we used the first frame of the Ground-Truth video as the condition image input. For EchoDiffusion and EchoNet-Synthetic, we provided the EF value of the Ground-Truth video as input, whereas for EchoPulse, the ECG signal of the Ground-Truth video was used. Quantitative evaluation metrics were then calculated for the generated and Ground-Truth video pairs. Additionally, we compared the EF values extracted from the generated videos with those of the Ground-Truth videos. As it is shown in Table 3. Table 3 further demonstrates the advantage of using ECG as a condition in our model by calculating the deviations of ED and ES key frames between the generated videos and the ground-truth videos.

Regarding sampling time, we adhered to the original EchoDiffusion setup and calculated the time required to generate 64 frames, corresponding to 8 complete ECG cycles, or eight heartbeats. EchoDiffusion and EchoNet-Synthetic were similarly adjusted to generate 64 frames, ensuring the output covered 8 heartbeats. It is worth noting that EchoNet-Synthetic employs a video editing-like approach, where the 64-frame video is generated by randomly repeating a single heartbeat cycle. This differs fundamentally from EchoPulse, which generates videos with higher information content. Direct comparisons under these conditions would be unfair. Therefore, in this experiment, we modified the generation length of EchoNet-Synthetic to match that of EchoPulse by generating eight heartbeat cycles in separate steps using the same EF value. This adjustment allowed for a fair comparison of generation speed.

## A.3 DATASET

Our private dataset consists of 94,078 ECHO videos, including both apical 2 view and apical 4 view. It contains ECHO recordings from both healthy individuals and patients with various cardiovascular diseases, with real-time ECG signals recorded for each video. All videos were anonymized, underwent random contrast enhancement, and were resized to 128x128 resolution.

Electrocardiography (ECG) is the most direct and reliable signal for capturing and analyzing the electrical activity of the heart, which is essential for understanding cardiac function. The ECG signal is generated by the depolarization and repolarization of myocardial cells during the cardiac cycle. This electrical activity originates in the sinoatrial (SA) node, often referred to as the heart's natural pacemaker, which initiates an electrical impulse. The impulse propagates through the atria, causing atrial contraction, and is then delayed at the atrioventricular (AV) node to allow proper

Table 7: **Prediction errors for ED and ES frames**, calculated by comparing the segmented frames from the ground-truth video and the generated video. These errors validate the consistency of cardiac motion in the generated video and provide insights into whether the model accurately follows ECG-driven dynamics.

| Methods | Average Error (ED)↓ | Average Error (ES)↓ |
|---|---|---|
| EchoDiffusion (Reynaud et al., 2023) | 1.55 | 1.25 |
| EchoNet-Synthetic(Reynaud et al., 2024) | 1.06 | 0.97 |
| ECHOPulse (Our Model) | 0.18 | 0.21 |

ventricular filling. It subsequently travels through the bundle of His, bundle branches, and Purkinje fibers, leading to coordinated ventricular contraction. The resulting electrical activity produces the characteristic waveforms observed in an ECG: the P wave (atrial depolarization), the QRS complex (ventricular depolarization), and the T wave (ventricular repolarization).

Ejection Fraction (EF) is a critical parameter for evaluating cardiac function, representing the percentage of blood volume ejected from the left ventricle during systole relative to the total volume at end-diastole. It is calculated using the left ventricular volumes at two key phases of the cardiac cycle: end-diastole (EDV/LVED) and end-systole (ESV/LVES). The general formula for EF is expressed as:

$$EF = \frac{\text{EDV} - \text{ESV}}{\text{EDV}} \times 100\%$$

or

$$EF = \frac{\text{LVED} - \text{LVES}}{\text{LVED}} \times 100\%$$

## A.4    ADDITIONAL GENERATED SAMPLES AND ANALYSIS

We asked one experienced ECHO clinician with over 10 years of expertise and a postdoctoral researcher with 8 years of experience in ECHO engineering to manually evaluate 100 video samples. Among these, 50 were generated samples, and 50 were true samples from the original dataset. The evaluators were tasked with classifying each video as real or fake. The expert accuracies were 51.17% and 49.39%, respectively. Both experts reported that the generated videos were nearly indistinguishable from the real ones.

Additional video generation examples are shown in the Figure 4,5,6 and 7. The comparison among ECHOPulse, EchoDiffusion and EchoNet-Synthetic examples can be seen in Figure 8 and 9. Both EchoDiffusion and EchoNet-Synthetic exhibit issues such as missing details, inaccuracies in valve structures, and incorrect valve motion phases. In contrast, ECHOPulse demonstrates significant improvements in these aspects, highlighting the advantages of using ECG as a control condition. However, ECHOPulse still has failure cases under extreme conditions, such as poor-quality condition images and ECG signals that deviate from normal pattern, as shown in Figure 10. Visualization results of model's reconstruction results are in Figure 11.

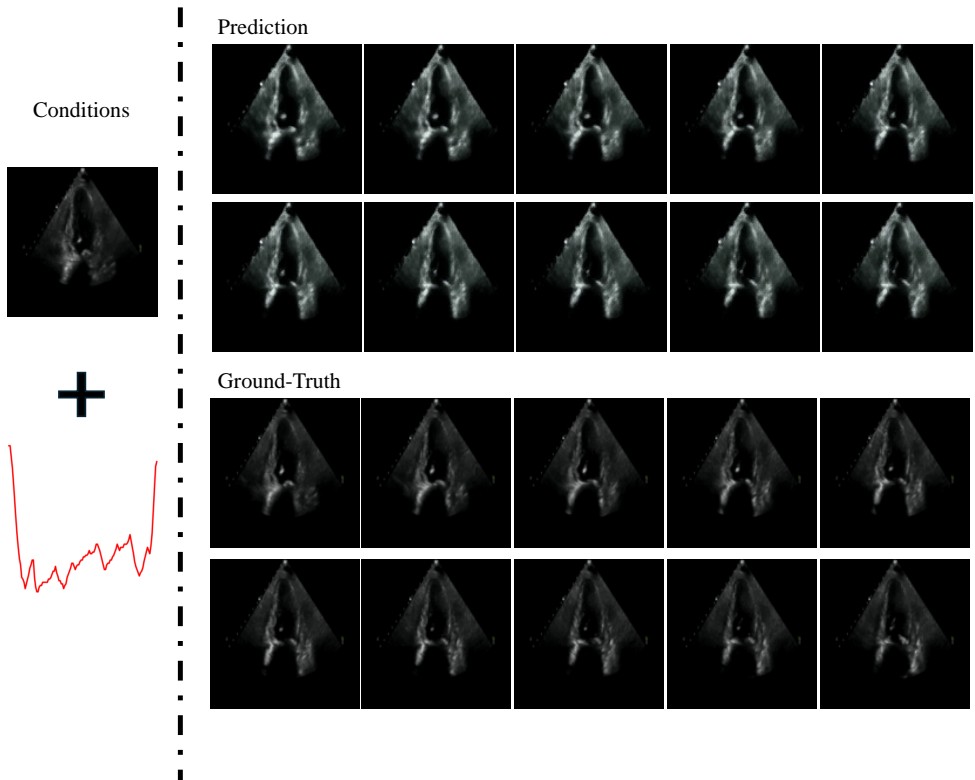

Figure 4: A2C video generation example from ECHOPulse.

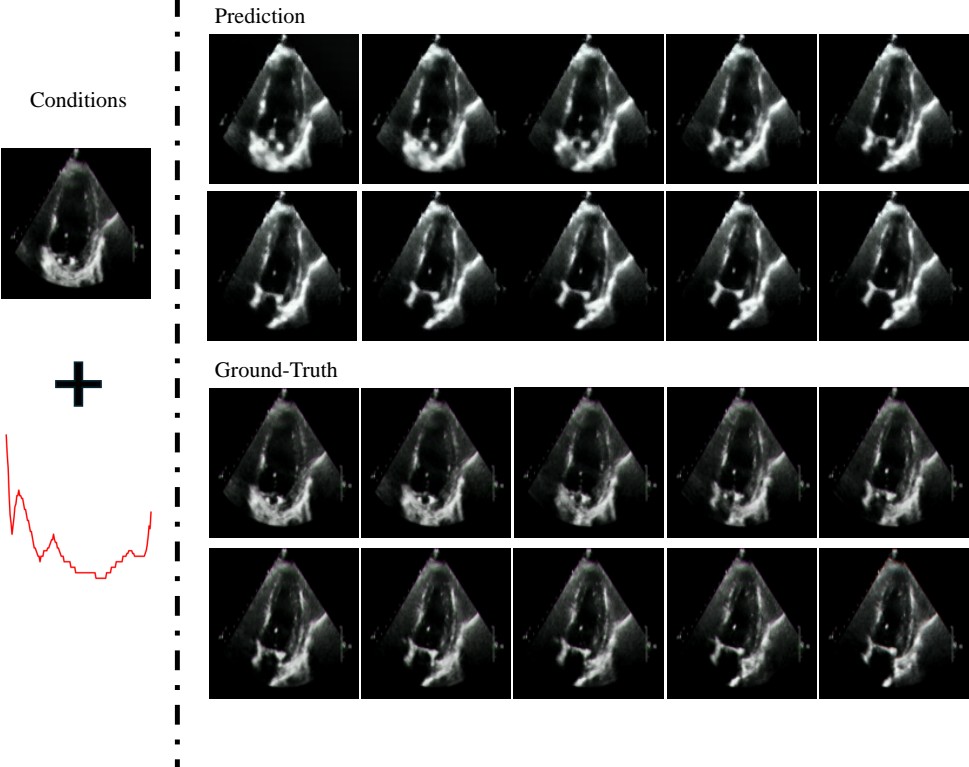

Figure 5: A2C video generation example from ECHOPulse.

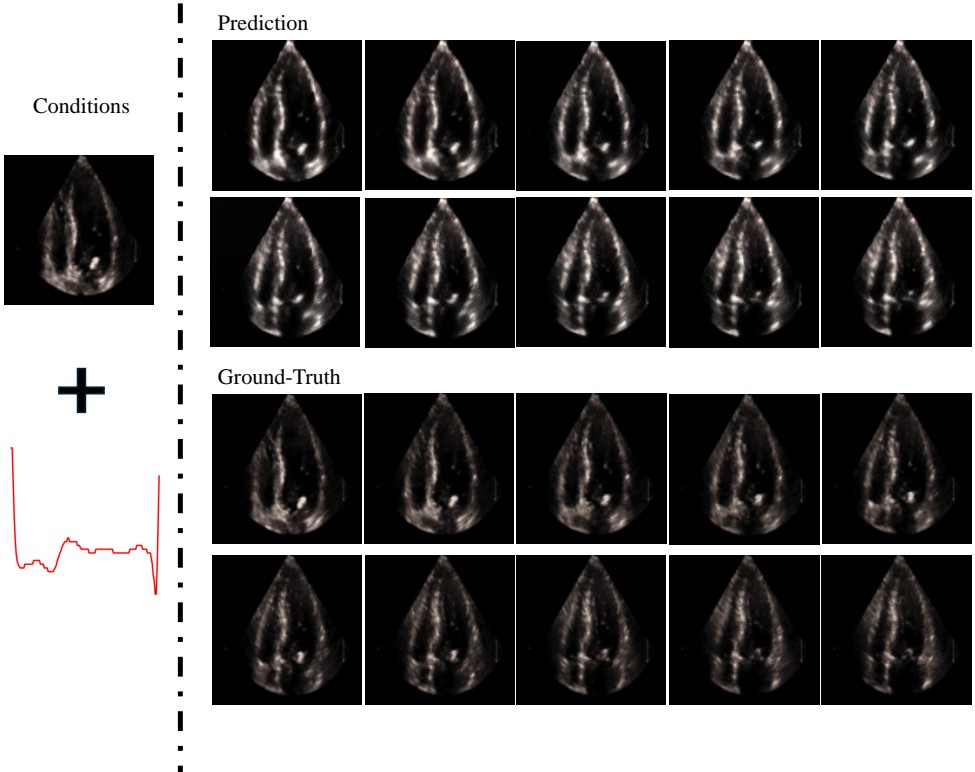

Figure 6: A4C video generation example from ECHOPulse.

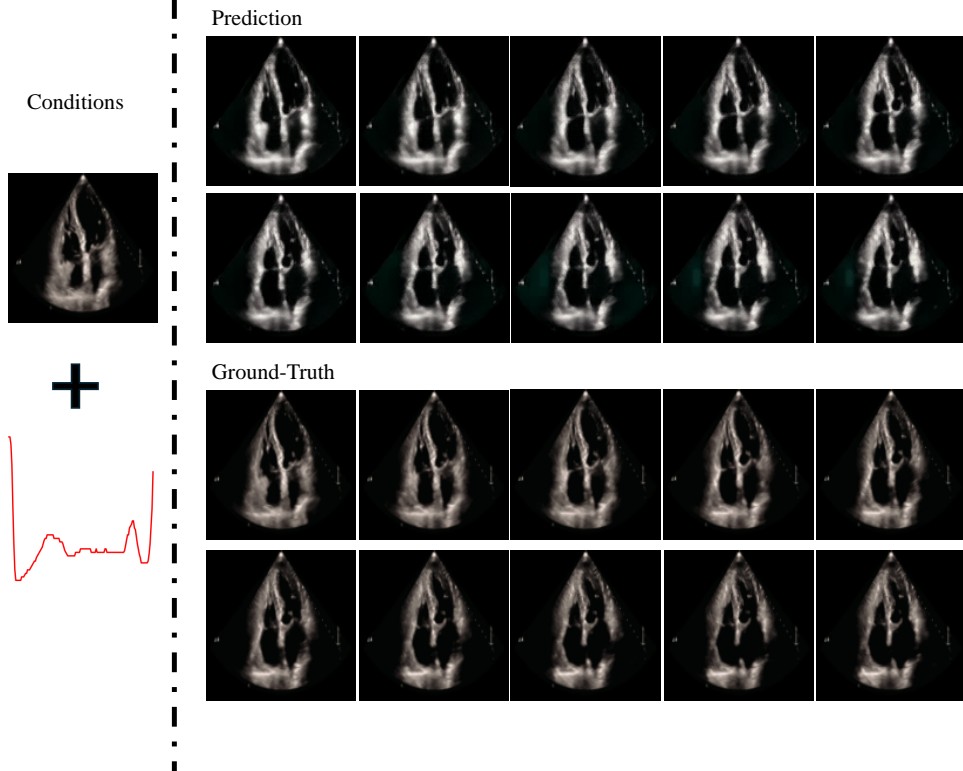

Figure 7: A4C video generation example from ECHOPulse.

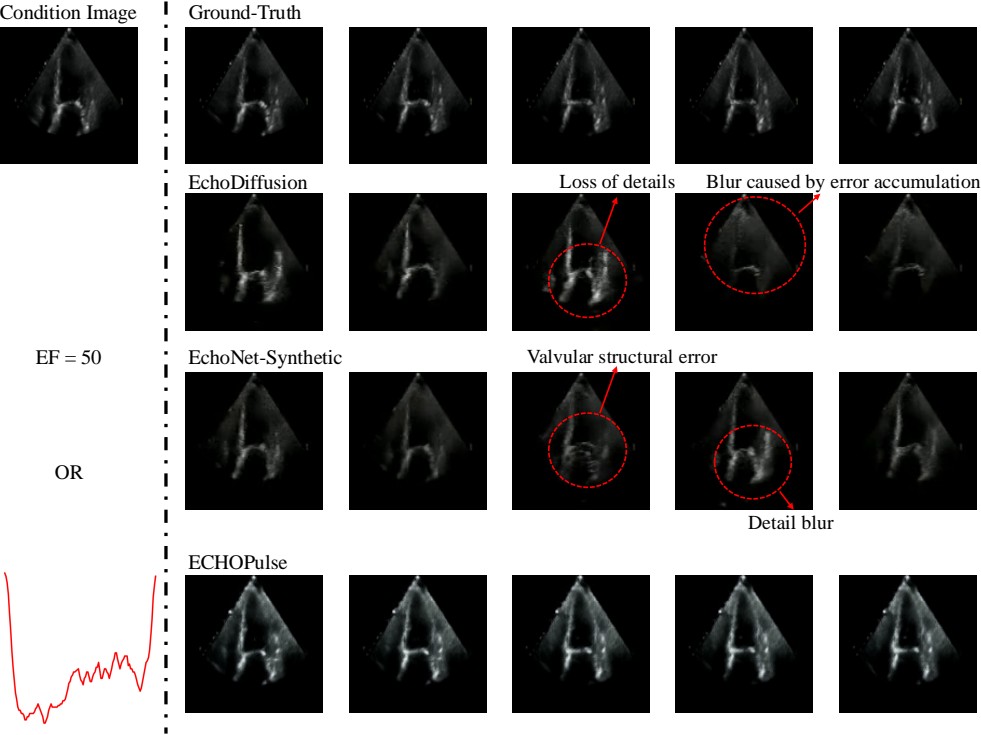

Figure 8: A2C video generation comparison results among ECHOPulse, EchoDiffusion and EchoNet-Synthetic.

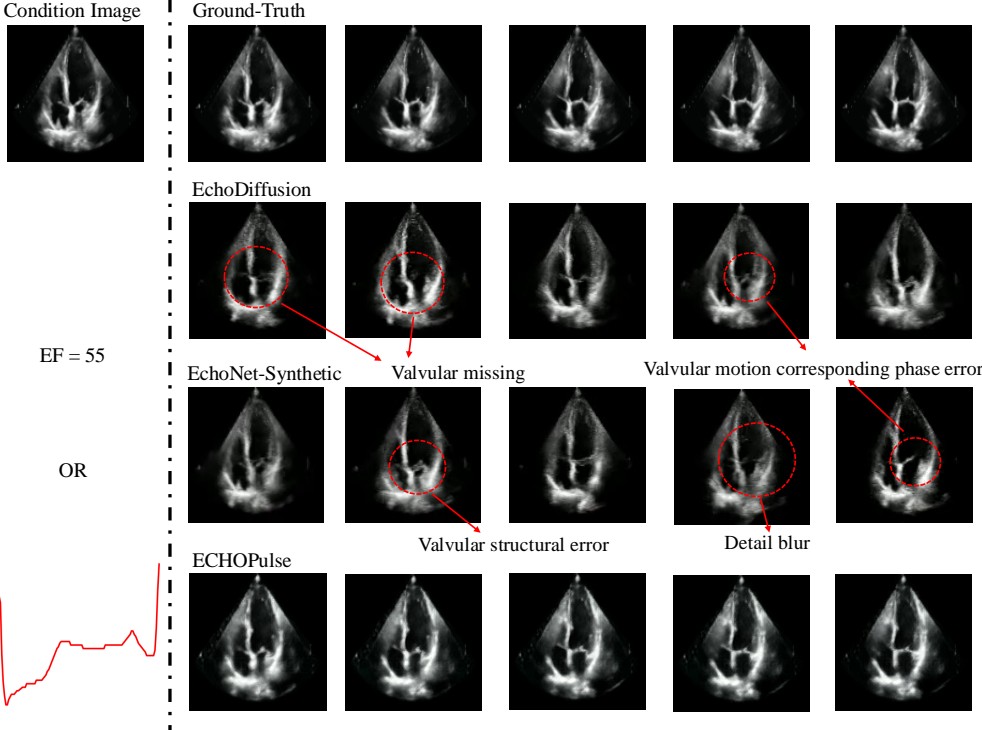

Figure 9: A4C video generation comparison results among ECHOPulse, EchoDiffusion and EchoNet-Synthetic.

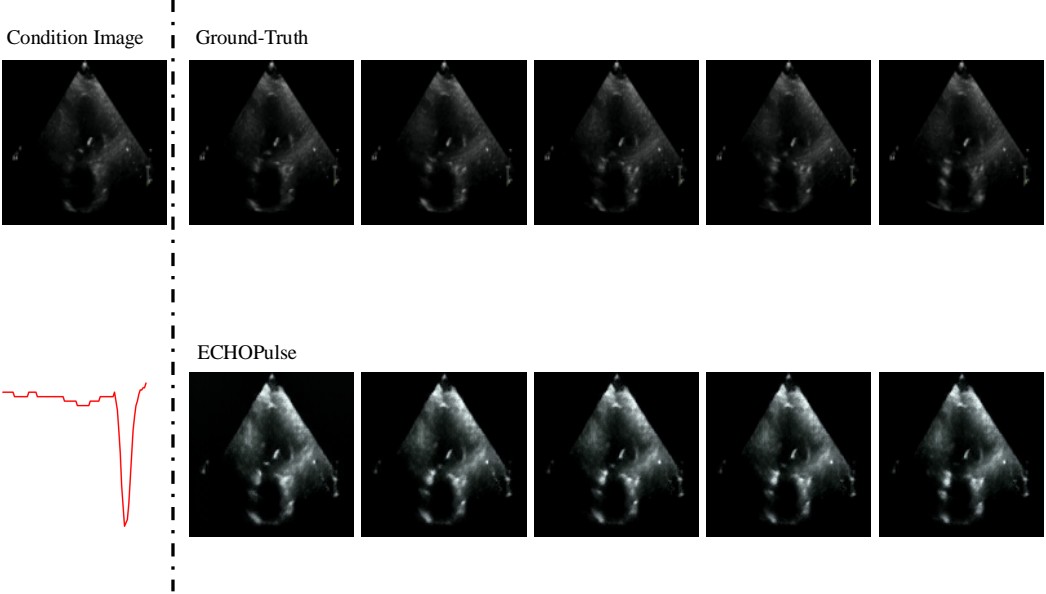

Figure 10: ECHOPulse's failure case occurs when encountering extremely poor-quality condition images and abnormal ECG patterns that deviate from typical physiological norms.

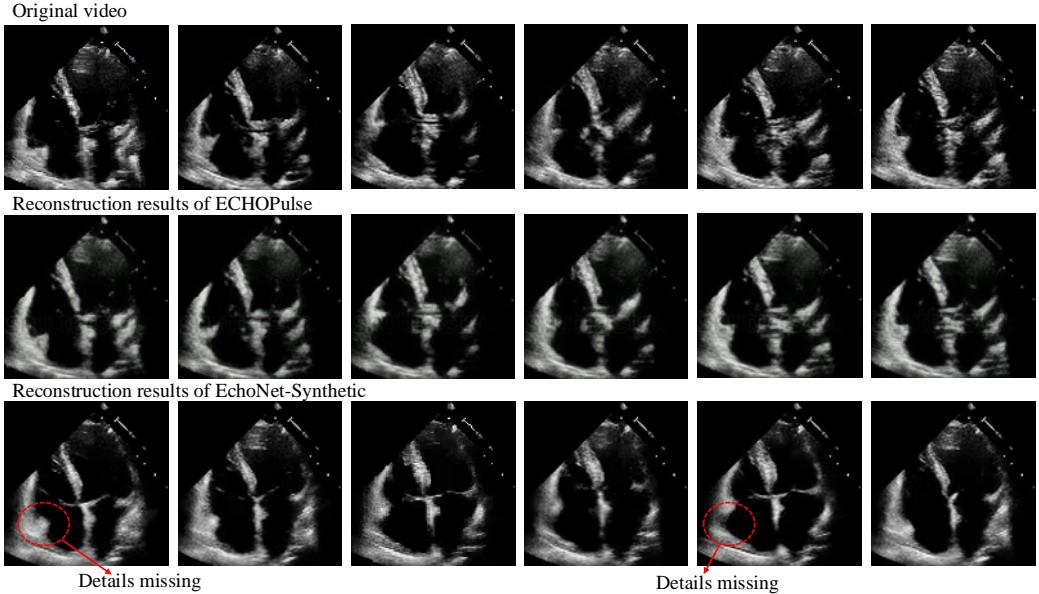

Figure 11: Reconstruction results comparison between ECHOPulse and EchoNet-Synthetic.

