# OpenReview forum: "ECHOPulse: ECG Controlled Echocardio-gram Video Generation"
_ICLR.cc/2025/Conference — ICLR 2025 Poster_

### Official Review · Reviewer_F9gf · 2024-10-19

**Soundness:** 3
**Presentation:** 3
**Contribution:** 2
**Rating:** 8
**Confidence:** 5

**Summary:**

The authors propose a Transformer and VQVAE-based generative model for echocardiogram generation, conditioned on ECG signals. This is a novel approach that seeks to explore ECG-based conditioning for controlling cardiac motion during echocardiogram generation, a relatively under-explored area.

**Strengths:**

- The use of ECG signals for controlling cardiac motion in echocardiogram generation is a novel and under-explored area, making the concept innovative and potentially impactful.
- The methodology is well-grounded in leveraging VQVAE tokenization for fast decoding, which addresses some of the computational bottlenecks in generative models.

**Weaknesses:**

several concerns and inaccuracies in the paper need to be addressed for clarity and rigor.

First, the abstract claims that existing methods have high computational costs, slow inference, and rely on complex conditioning prompts. This statement is not entirely accurate. For example, the paper does not provide a comparison with EchoNet-Synthetic when discussing generation times, which is essential for validating such claims. Neglecting such a comparison is poor practice and can mislead the reader about the novelty and efficiency of the proposed method.

In Section 2.3, there is a reference to "Reynaud et al. (2024) introducing EchoNet-Dynamic." However, this passage is unclear, and it appears there might be confusion between the works of Reynaud et al. (2024) and Ouyang et al. (2020). Clarifying this would help avoid misattribution of prior work.

Additionally, the statement that these studies are conditioned on carefully curated prompts such as segmentation masks and clinical text is not fully correct. Only HeartBeat (Zhou et al., 2024) uses these forms of conditioning. EchoDiffusion (Reynaud et al., 2023) and EchoNet-Synthetic (Reynaud et al., 2024) use image and EF as conditioning inputs. This oversimplification of related work needs to be corrected.

Furthermore, Eq. 1 lacks proper citations. The loss function used in this equation appears to be inspired by previous works, yet no references are given. The authors should consider citing relevant works such as Gu et al. (2022) [1], Rombach et al. (2022) [2], and Esser et al. (2021) [3] to give due credit.

Section 4.1 presents an incorrect conclusion. The numbers extracted from the EchoNet-Synthetic paper reflect only the VAE model metrics, without involving the diffusion model. Similarly, the metrics for ECHOPulse involve only the VQVAE, not the token-prediction model. Therefore, the comparison between VQ-VAEs and diffusion models is not valid, and the conclusion that VQ-VAEs outperform diffusion models cannot be drawn from this comparison.

In Table 2, there are several points that need further explanation. It is unclear how MoonShot and VideoComposer were evaluated for this task, as the numbers seem to have been taken from the HeartBeat paper without further elaboration. Additionally, EchoNet-Dynamic includes only A4C views, so it is not clear how ECHOPulse was evaluated on A2C views, given that these metrics are not reported in EchoDiffusion and EchoNet-Synthetic. The SSIM computation is also problematic, as SSIM generally requires paired input/output data, and it is not clear how ECHOPulse reconstructs a video. Moreover, how the text conditionings for CAMUS and EchoNet-Dynamic were generated should be clarified. Lastly, the table construction raises questions, particularly why HeartBeat and EchoNet-Synthetic are singled out at the end.

Section 4.2 discusses the comparison with baselines, but the experimental setup lacks sufficient explanation, making the comparisons appear unfair. MoonShot and VideoComposer are general -- purpose generative models, so it is unsurprising that they perform worse than domain -- specific models. HeartBeat’s use of multiple conditions emphasizes the importance of using as many conditioning variables as possible. However, only the ablation results of HeartBeat are compared to other methods, which limits the value of the comparison. Furthermore, the conclusion of this section is poorly formulated. While the qualitative results clearly indicate that the visual quality of ECHOPulse samples lags behind, the conclusion only hints at this issue without directly acknowledging it. Moreover, the FID and FVD metrics are poor indicators of quality in this specific task, and this should be discussed more critically.

In Table 3, EchoNet-Synthetic is conspicuously absent. Including this model would provide a more complete comparative analysis.

In the paragraph following Table 3, the method for calculating LVEF in this paper is substantially different from that used for the EchoNet-Dynamic dataset, making direct comparisons problematic. This discrepancy is not adequately addressed, and the conclusion is again misleading, as EchoNet-Synthetic is not included in this analysis, even though it shows superior performance in all reported metrics.

Section 5 makes some strong claims about the performance of the proposed approach. While the novelty of the idea is clear, the assertion that it achieves the best results is far-fetched. Previous works such as EchoNet-Synthetic and HeartBeat have demonstrated better performance. The authors should focus on highlighting the innovative aspects of their method rather than making overstated claims about its superiority.

Overall, using ECG signals for ECHO video generation is an intriguing approach. The idea of using ECG is interesting, the method (VQVAE + Transformer) is less common in the field if image/video genetation, similar to Phenaki [4]. However, the literature review and baseline comparisons are inadequate, and the paper is riddled with oversights and false conclusions. A more thorough review and revision of the comparisons and claims would significantly improve the work. Additionally, proper citations for key components of the methodology are necessary.

References:
[1] Gu, S., Chen, D., Bao, J., Wen, F., Zhang, B., Chen, D., Yuan, L., and Guo, B., 2022. Vector quantized diffusion model for text-to-image synthesis. In Proceedings of the IEEE/CVF conference on computer vision and pattern recognition (pp. 10696-10706).
[2] Rombach, R., Blattmann, A., Lorenz, D., Esser, P. and Ommer, B., 2022. High-resolution image synthesis with latent diffusion models. In Proceedings of the IEEE/CVF conference on computer vision and pattern recognition (pp. 10684-10695).
[3] Esser, P., Rombach, R. and Ommer, B., 2021. Taming transformers for high-resolution image synthesis. In Proceedings of the IEEE/CVF conference on computer vision and pattern recognition (pp. 12873-12883).
[4] Villegas, R., Babaeizadeh, M., Kindermans, P.J., Moraldo, H., Zhang, H., Saffar, M.T., Castro, S., Kunze, J., and Erhan, D., 2022, October. Phenaki: Variable length video generation from open domain textual descriptions. In International Conference on Learning Representations.

**Questions:**

- Can you provide specific details or comparisons that validate the claims in the abstract regarding the computational cost, slow inference, and complex conditioning of existing methods? Why was EchoNet-Synthetic not included when discussing generation times?

- There appears to be some confusion between the works of Reynaud et al. (2024) and Ouyang et al. (2020) in Section 2.3. Could you clarify the differences between these works and how they relate to your methodology?

- The paper suggests that prior works such as EchoDiffusion and EchoNet-Synthetic use complex conditional prompts like segmentation masks and clinical text, but these models actually use image and EF for conditioning. Could you revise this section to more accurately reflect the conditioning methods used in these studies?

- The loss function presented in Equation 1 lacks references to foundational works. Could you cite the relevant literature that inspired or derived this loss, such as Gu et al. (2022), Rombach et al. (2022), or Esser et al. (2021)?

- The comparison between VQ-VAE and diffusion models seems incomplete, as the EchoNet-Synthetic metrics presented relate only to the VAE model. Could you clarify how these models were compared and whether it is appropriate to conclude that VQ-VAEs surpass diffusion models based on these metrics?

- How were MoonShot and VideoComposer evaluated for this specific task? Did you replicate their results, or were the metrics taken from the HeartBeat paper? Additionally, can you clarify how SSIM was computed, considering it typically requires paired input/output data, which is not evident for ECHOPulse?

- EchoNet-Dynamic contains only A4C views. How was ECHOPulse evaluated on A2C views, given that these metrics are not reported in EchoDiffusion and EchoNet-Synthetic?

- The comparison with baselines such as MoonShot and VideoComposer appears unfair, given that these are general-purpose models. Could you explain how the experimental setup ensures fair comparison with domain-optimized models, and why only the ablation results of HeartBeat were compared?

- EchoNet-Synthetic is conspicuously absent from Table 3. Why was this model excluded from the comparison, and how would including it impact the conclusions?

- The method for calculating LVEF in this paper differs significantly from that used in the EchoNet-Dynamic dataset. How do you account for this discrepancy, and does it affect the validity of the comparisons?

- While the novelty of using ECG signals is appreciated, the claim that your model achieves state-of-the-art results seems overstated. Can you clarify the specific advantages of your model compared to EchoNet-Synthetic and HeartBeat, without overstating the performance?

- The qualitative results suggest that the visual quality of ECHOPulse samples is lower than that of competing models, but this is not explicitly acknowledged in the text. How do you address this issue, and why were FID and FVD metrics chosen for evaluation, given their limitations for this task?

---

> ### Author Response · Authors · 2024-11-21
>
> We sincerely thank you for your thoughtful and constructive feedback, which has been invaluable in improving our work and clarifying its contributions. Your insights have provided important perspectives, and we are committed to addressing them to enhance the quality of our manuscript. We have submitted the revised manuscript, with all revisions highlighted in blue. The weaknesses you mentioned are also addressed in the Questions section. Due to space limitations, we have responded to them all within that part of the paper.
>
> **Question Response:**
>
> **[Q1]Computational Cost:**  Regarding EchoNet-Synthetic, we did not include its inference time in the original paper because our reproduction on our private dataset showed that ECHOsynthetic takes approximately 24 seconds to generate 64 frames on a single A100 80G, likely due to differences in inference methods. Since this result deviates from their reported performance, we opted not to include it. We have included this result in the revised version of the table and details of the inference time calculation methods in the appendix.
>
> Additionally, we feel that EchoNet-Synthetic is essentially a video editing method. Its input is essentially a duplicated image sequence that undergoes video editing with VAE decoding. Like our model, ECHOsynthetic benefits from VAE's decoding speed. However, their method generates 64 frames (three heartbeats) based on a single EF value under diffusion randomness, while our model generates 64 frames corresponding to eight complete ECG cycles, representing eight heartbeats. This significant difference in information density and prediction difficulty makes direct speed comparisons somewhat unfair.
>
> **[Q2]Typographical Error:**  It is really sorry that we made this mistake in Section 2.3 It should be EchoDiffusion, We sincerely apologize for the oversight caused by our typographical error. EchoDiffusion is a method based on diffusion to synthetic ECHO videos while EchoNet-Dynamic is the dataset. We have revised it in the newest version.
>
> **[Q3]Conditional Prompts:**  We apologize for any misunderstanding caused by our wording. When we mentioned methods requiring a large number of complex conditions, we were specifically referring to HeartBeat. In contrast, EchoDiffusion and EchoNet-Synthetic only require an image and EF value as inputs. We have revised our description to make this distinction clearer and more accurate.
>
> That said, it is worth noting that EF is itself a complex metric. Calculating EF requires deriving ED and ES areas from the patient’s heart video, which involves a detailed and intricate process. Thus, EF can also be considered a condition obtained through a complex procedure, often required experts’ annotation, which is also the reason why their model can only work for video synthetic while ours have the potential for clinical application. Thank you for bringing this to our attention.
>
> **[Q4]Citation:**  We have added the correct citation. Thank you very much for correcting us.
>
> **[Q5]Comparison between VAE and Diffusion:** Thank you for pointing this out. We have revised this part of the description to compare only the VAE component of EchoNet-Synthetic with our model. The results have been updated in the revised manuscripts and the reproducing details can be found in Appendix.  Our intention is not to claim that our VQVAE outperforms diffusion models but rather to demonstrate that our VQVAE training was successful. The discretization process did not result in significant information loss, which validates the effectiveness of our approach. Our VQVAE did not beat the diffusion model, we only defeated the EchoNet-Synthetic’s VAE in this task. We have revised the wordings to make it more accurate.  We really appreciate your feedback.
>
> **[Q6]Video Generation Comparison:** Moonshot and VideoComposer directly use the results reported in the HeartBeat paper, as these two general-purpose models were not retrained. However, since the original HeartBeat paper does not provide sufficient implementation details and has not released its source code, we did not conduct comparisons on our private dataset. Instead, we supplemented our experiments with EchoDiffusion and EchoNet-Synthetic, according to your suggestions. Results can be found in the revised version.
>
> Our private dataset includes ECG data, with each video corresponding to an ECG signal. This alignment allows the condition image-generated videos to have corresponding ground truth videos, enabling us to compute SSIM as well. Thank you for your understanding, and we hope this clarification addresses your concerns.

---

> > ### Author Response · Authors · 2024-11-21
> >
> > **[Q7] A2C and A4C Issue:**  We sincerely apologize for this significant error. For the task of video generation, it is not necessary to separate A2C and A4C views in ECHO. During training, we did not make this distinction. However, to maintain consistency with the HeartBeat baseline, we trained a separate classification model to distinguish A2C and A4C views within our private dataset purely for the purpose of comparison.
> >
> > That said, some A4C videos are visually similar to A2C, leading our model to occasionally make incorrect classifications. We failed to recognize that the EchoNet-synthetic dataset contains only A4C videos and mistakenly applied our classification model in this dataset, causing a misunderstanding. We deeply regret this error.
> >
> > We have recalculated the results and updated them in the table accordingly. Thank you for your understanding, and we will take greater care in ensuring the accuracy of our analysis in the future.
> >
> > **[Q8]Absent EchoNet-synthetic from Table 3:**  According to your suggestions, we have reproduced EchoDiffusion and EchoNet-Synthetic on our private dataset. The comparison results have been added to Table3 and Table2. The answer to this question is similar to **Q1**.
> >
> > **[Q9] LVEF Calculation:** This was indeed an oversight on our part, and we have provided a clearer explanation in the Appendix. The equation used by EchoNet-Dynamic is actually the same as ours. EDV can also be written as LVED, ESV represents LVES. Thank you for bringing this to our attention. Thank you very much for your thorough review. We are honored that our paper has been evaluated by such a responsible reviewer like you.
> >
> > **[Q10]Overstatement Issue:** The greatest strength of our model lies in its use of ECG as a condition for video generation. ECG inherently contains EF information (albeit with some error) as well as temporal signals, making it exceptionally well-suited for this task. The success of our video generation is largely attributable to this choice.
> >
> > In contrast, the main limitation of HeartBeat and EchoNet-synthetic is that their conditions are less readily obtainable. EF values, for instance, require segmentation results from video, which is not a straightforward process. By comparison, our model offers greater practical application potential.
> >
> > We fully understand that the video quality generated by VQVAE may not match that of diffusion models. However, both approaches demand considerable engineering effort. A well-trained VQVAE can achieve performance comparable to diffusion models. We also want to directly compare with VQVAE-based models such as MAGVIT[2] and MAGVIT2[1], but their codes and weights are not open source, and we do not have such large computing resources to reproduce their results. Under these conditions, the advantages of our video generation model become apparent. Thanks to its LLM-inspired training methodology, our model offers faster generation speeds, flexibility in generation length, and potential compatibility with foundation models in the future. These attributes highlight its versatility and potential for broader applications.
> >
> > **[Q11]Qualitative Results:** According to the reviews’ suggestions, we have reproduced EchoDiffusion and EchoNet-Synthetic and updated the results in the revised paper. Our model does not produce lower-quality videos compared to other models, as evidenced by its performance on the private dataset. The lower results on the CAMUS dataset can be attributed to HeartBeat using seven conditions, while our model relies solely on textual input. Similarly, on the EchoNet-Dynamic dataset, our performance appears lower because EF values were provided to our model as text inputs. Since EF encoding was not the primary focus of our study, we did not design a specialized numerical encoder for EF, which explains why our model falls behind in comparisons under these conditions.
> >
> > As for FID and FVD metrics, we fully understand that they are not ideally suited for this task. However, to enable a fair comparison with existing methods, we calculated these metrics as well. That said, metrics like EF estimation, which better reflect the practical performance of our model, are a more accurate representation of its capabilities. And according to other reviewer’s comments, we have put more comparison examples in the Appendix, which can help visualize our model’s better performance. These help evaluation clearly demonstrate the strengths of our approach.

---

> > > ### Author Response · Authors · 2024-11-21
> > >
> > > Finally, I would like to sincerely thank you for the time and effort you have dedicated to reviewing our paper. You are the most meticulous reviewer I have encountered so far. We deeply appreciate the time you spent analyzing the methods and datasets cited in our paper, as well as the detailed and thoughtful questions you raised.  If it were possible to rate reviewers, I would undoubtedly give you a perfect score of 10. Your efforts have significantly contributed to advancing our research. Thank you very much.
> > >
> > > **Reference**
> > >
> > > [1] Yu, Lijun, et al. "Language Model Beats Diffusion--Tokenizer is Key to Visual Generation." arXiv preprint arXiv:2310.05737 (2023).
> > > [2] Yu, Lijun, et al. "Magvit: Masked generative video transformer." Proceedings of the IEEE/CVF Conference on Computer Vision and Pattern Recognition. 2023.

---

### Official Review · Reviewer_vh23 · 2024-10-29

**Soundness:** 2
**Presentation:** 3
**Contribution:** 2
**Rating:** 6
**Confidence:** 4

**Summary:**

The authors propose Echopulse, a video diffusion model conditioned on time-series data. Their pipeline is split into different parts including a tokenization step for efficient generation, a feature alignment step to align visual features with time series data, and a generation step for video generation.
In general, the paper is well written and easy to follow.
While their contribution is clear, I am not convinced that the proposed method is better than what was previously proposed in the literature. The main experiments lack exhaustive comparison (eg. Tab.1 private data, tab2 private data) and multiple things proposed in the paper are not properly validated. Overall I give a weak reject.

**Strengths:**

I believe the idea of aligning the representation of frames with ECG data is very good and has a large potential to work for other domains as well.

The writing is very clear and from beginning to end it is easy to understand what the authors are doing.

I believe that conditioning videos on time-series data could have a strong impact even outside of ultrasound videos.

**Weaknesses:**

There is not enough evidence supporting the claim that the generated videos follow the ECG signal. The LVEF estimation presented in Table 3 only considers ED and ES phases which are only a part of the ECG cycle. Furthermore, the results on EF estimation are inferior compared to previous methods in terms of R2 score which is the gold standard metric for estimating EF. Figure 3 only shows one example and is difficult to understand. Maybe it would help to add LVED and LVES values to the frames.

Table 3 is missing references. Why is EchoNet-Synthetic not mentioned as a baseline in Table 3? Their inference time is below 3 seconds.

It is unclear how much influence reconstruction has on the final results. While the reconstruction results of the method presented are best, I am not convinced that the difference will be visible when using the method for downstream tasks such as video generation or EF estimation. I think it would be helpful to provide metrics such as rFID or rLVEF (reconstruction LVEF) to get an estimate to how much information is lost through encoding and how much better EchoPulse is compared to previous methods.

Inconsequent comparison: Why are there no generative results for EchoDiffusion or EchoNetSynthetic on Private data? Also, Table 1 misses the EchoNet-Synthetic baseline.

Motivation: While I think the application is exciting I believe there is a lack of motivation for ECG to video generation in particular. It would help if you could further motivate. Especially on what is the motivation of doing ECG conditioned generation over EF condtioined video generation.

Minor:
Table 2 should be separated into all datasets to make comparison easier.

**Questions:**

Do you have details about how good the models are without pretraining on natural data?

---

> ### Author Response · Authors · 2024-11-21
> **Part1**
>
> We sincerely thank you for your thoughtful and constructive feedback, which has been invaluable in improving our work and clarifying its contributions. Your insights have provided important perspectives, and we are committed to addressing them to enhance the quality of our manuscript. We have submitted the revised manuscript, with all revisions highlighted in blue.
>
> **Weakness Response:**
>
> **[W1]Quality Evaluation:**  Thank you for your valuable suggestions. We realize that a single figure, like Figure 3, is insufficient to fully explain our results. We have added more examples in the Appendix. We have revised the annotations and provide more detailed explanations of the relevant medical concepts in the Appendix. The intention of Figure 3 was to illustrate that ED and ES phases correctly align with the corresponding ECG phases in the generated video frames. As the ECG signal changes, the positions of ED and ES also shift accordingly. Metrics such as FID and FVD also provide supporting evidence of this alignment.
> Regarding the slightly lower R² compared to ECHOdiffusion, this can be attributed to the differences in training datasets. ECHOdiffusion was trained exclusively on a dataset of four-chamber heart views with relatively high-quality data and minimal EF variability. In contrast, our dataset is over nine times larger, including diverse conditions that introduce more EF variability.  According to reviewer’s comets, we have reproduced EchoDiffusion and EchoNet-Synthetic on our private datasets and revised the metrics in the newest version of paper. The results shows that our model has a better performance in all aspect.
>
> **[W2]Table 3:**  Regarding EchoNet-Synthetic, we did not include its inference time in the original paper because our reproduction on our private dataset showed that EchoNet-Synthetic takes approximately 24 seconds to generate 64 frames on a single A100 80G, likely due to differences in inference methods. Since this result deviates from their reported performance, we opted not to include it. However, we have included this result in the revised version of the table and detail the inference time calculation methods in the appendix.
> Additionally, we feel that EchoNet-Synthetic is essentially a video editing method. Its input is essentially a duplicated image sequence that undergoes video editing with VAE decoding. Like our model, EchoNet-Synthetic benefits from VAE's decoding speed. However, their method generates 64 frames (three heartbeats) based on a single EF value under diffusion randomness, while our model generates 64 frames corresponding to eight complete ECG cycles, representing eight heartbeats. This significant difference in information density and prediction difficulty makes direct speed comparisons somewhat unfair.
>
> **[W3]Reconstruction influence:** We are grateful for your critique and have provided the rFID and rLVEF results, which we have included in the Appendix. As noted in the paper, VQVAE discretizes variables, leading to some loss of information compared to continuous-variable models like VAE. However, according to the compression ratio concept outlined in the VQGAN paper, our selected codebook size (8192) and video patch size (8×8) yield a compression ratio of 19.69 for videos with a resolution of 128×128. Empirical experience suggests that compression ratios between 4 and 32 generally do not significantly affect video generation quality. Our parameter choices were based on these guidelines.
>
> Our intention with Table 1 was not to claim that our method outperforms EchoNet-Synthetic’s VAE in reconstruction but rather to demonstrate that our discrete VAE achieves comparable performance to continuous VAE models, validating our video generation capability. The discrete representation was chosen to leverage the advantages of LLM-like training and prediction, at the cost of some video information loss. Table 1 sufficiently demonstrates that our method does not lose more information than existing ECHO generation models.
>
> We have also provided EchoNet-Synthetic’s reconstruction performance on our private dataset. It is worth noting that our VQVAE was pretrained on natural images and fine-tuned on the CAMUS dataset (500 high-quality videos) rather than trained on private data. This decision was made to avoid learning potentially sensitive patient-specific features, ensuring privacy preservation. Despite this, our VQVAE still achieves commendable reconstruction performance.
>
> In contrast, EchoNet-Synthetic’s VAE exhibited weaker generalization and required retraining on our private dataset to achieve performance similar to their original paper. This raises concerns about privacy, as their VAE was trained on private datasets despite their acknowledgment of privacy issues. If we were to train our VQVAE on private data, the reconstruction metrics would likely improve further, but privacy concerns remain a priority for us.

---

> > ### Comment · Reviewer_vh23 · 2024-11-25
> >
> > Re W1: Observing the generated videos in the supplements I am not entirely convinced that the generated videos accurately follow the input ECG signal. In A.2.3 you mention that you "compared the EF values extracted from the generated videos with those of the Ground-Truth videos." is that the rLVEF in Table 6?
> > Additionally, from looking at the generated samples it looks like that the generated samples not only follow the ECG signal but also the anatomy of the ground-truth video behind the ECG signal. Further, the apple watch example in the appendix lacks temporal consistency and resembles videos stitched to each other rather than longer synthetic videos.
> >
> > Re W2: I disagree with the assertion that EchoNet Synthetic is a video editing approach. Video editing starts from a real video and merely edits the video, whereas EchoNet Synthetic starts from random Gaussian noise like any other method.
> >
> > Re W3: Thank you for providing reconstruction scores. As of right now, table 6 is not mentioned in the text, neither did you explain what rLVEF means and you did not report FID and rFID. Please revise this table. I would appreciate it if you would add it to the main text. Since these are minor edits, they will not influence my final decision.

---

> > > ### Author Response · Authors · 2024-11-26
> > > **Further answers**
> > >
> > > Thank you again to read our rebuttal. Much appreciation.
> > >
> > > **Re W1**:
> > >
> > >  **1)** In the newly added Table 7, we calculate the frame differences (time discrepancies) between the ED and ES frames of the generated videos and the corresponding frames in the ground-truth videos. This aims to quantitatively evaluate the results. We acknowledge that it is challenging to visually demonstrate strict ECHO adherence to ECG motion within the paper. However, we have provided GIF examples in the supplementary Examples folder, which we hope will be helpful.
> > >
> > >  **2)** In Section A 2.3, we refer to Table 3, which compares the EF values of generated videos and ground-truth videos. Based on your suggestions, we reproduced EchoDiffusion and EchoNet-Synthetic on a private dataset and conducted comprehensive comparisons, adding and updating the original tables accordingly. We have modified this part in the paper. Additionally, Table 6 was introduced based on your feedback to include a comparison of reconstruction results, aiming to highlight information loss.
> > >
> > > **3)** We understand your concerns, and this is completely valid. Addressing the compounding factor issue raised by other reviewers as well, we emphasize that the examples we present are generated based on the condition image, effectively animating it. This ensures that the anatomical structure remains consistent, preserving the clinical relevance of the generated videos. Furthermore, our experiments reveal that having the model predict subsequent motion from the first frame is a significantly more challenging task compared to directly generating the video. Our model is also capable of generating videos without a condition image. If you are interested, I would be happy to provide demonstrations for you.
> > >
> > > **4)** Thank you very much for pointing this out. The perceived discontinuity can be attributed to two factors. First, it is partly due to the inherent characteristics of cardiac motion—cardiac contraction tends to be relatively slow, while relaxation and diastole occur much more rapidly. This trend is also evident in the ECG waveform. If you look at the ground truth GIFs provided in the supplementary materials, you can observe this pattern, which is entirely normal.
> > > Second, we are deeply grateful for your feedback on this matter. Since the Apple Watch data is entirely zero-shot for our model, it had not encountered such data during training. Additionally, the ECG signal was extracted from Apple device PDFs using OCR, which requires some engineering optimization to achieve the best performance. Each heartbeat’s duration exhibits slight variability, and during inference, I deliberately avoided normalizing the length of the signal, as I felt it might alter the characteristics of the ECG. Instead, I trimmed some of the extra ECG signal—namely, the flat portions—resulting in a slight reduction in the number of frames in the generated video. While this does not affect ED or ES frame detection, it is indeed a point for engineering improvement, and I take this feedback very seriously.
> > > I have updated the supplementary GIF files, the similar problems do not exist now.

---

> > > > ### Author Response · Authors · 2024-11-26
> > > > **Further answers Part2**
> > > >
> > > > **Re W2**: Thank you for pointing this out again. After careful review and discussion, we recognize the importance of this perspective. However, we want to clarify that this point was not emphasized in the manuscript but rather shared as a personal opinion during our discussion with the reviewer. Our observation stems from the nature of video generation in EchoNet-Synthetic, which appears to involve a degree of randomness and repetition. While its generation speed is indeed fast, it may not provide sufficiently strong control over the content in all cases. Thus, a direct comparison of generation speed alone might not be entirely fair. Nevertheless, your comment is valid, and we will revise the description accordingly.
> > > > Additionally, following the reviewer's suggestion, we replicated the results of EchoNet-Synthetic on a private dataset. Notably, the video resolution in the private dataset is 128 \times 128, whereas the EchoNet-Synthetic paper reported results for 112 \times 112 videos. In our comparative experiments, under the same experimental settings, our model demonstrated faster generation speeds (Table 3) and produced videos of higher quality than EchoNet-Synthetic. This reinforces the validity of our conclusions.
> > > >
> > > >
> > > > **Re W3**:  We are slightly concerned that making extensive changes to the main text at this stage might be flagged by the system as significant revisions to the manuscript. Therefore, we choose to put all the modifications in the Appendix at this stage. Nevertheless, we sincerely appreciate your feedback and will make the necessary adjustments.
> > > > As for Table 6, it currently presents the reconstruction results, **including Reconstruction FID (rFID) and reconstruction LVEF(rLVEF) as we mentioned to us. However, we are not entirely sure which "rFID" you are referring to**. Based on our current knowledge, we have encountered similar expressions **“Reconstruction FID (rFID)”** in other papers, but we have not come across any alternative calculation methods in the literature.  If possible, could you kindly clarify or provide further guidance? We will carefully address this feedback in our revisions.
> > > >
> > > > **Summary**
> > > > We deeply appreciate your efforts in pointing out the shortcomings of our work. We understand that your time is incredibly valuable, and we are truly grateful for your dedication and the time you’ve taken, even at the expense of your own work and rest, to help us improve the quality of our manuscript.
> > > >
> > > > You have provided insightful comments regarding W1, W2, and W3. We are wondering if you have any additional feedback on W4, W5, W6, or Q1? Based on your suggestions, we have added comparative experiments and conducted more in-depth analysis. Additionally, we are actively working on incorporating experiments for more downstream tasks.
> > > >
> > > >  We apologize for any inconvenience caused by reaching out again, but your suggestions have been immensely helpful to us. We would be sincerely grateful if you could kindly provide further feedback on these aspects. Thank you so much for your continued guidance.

---

> ### Author Response · Authors · 2024-11-21
> **Part2**
>
> **[W4] Inconsequent Comparison:** The results of EchoNet-Synthetic in Table 1 were provided in our previous response, and a comparative summary is included below. We reproduced two methods on our private dataset, and their metrics have been included in the revised version. It is evident that our model outperforms the others. As mentioned earlier,  EchoNet-Synthetic is essentially a video editing model, making direct comparisons with our generative model somewhat unfair.
>
> On our dataset, the performance of these two methods was not as strong as reported in their original papers. This discrepancy may be due to missing details in their training processes or codebases. Even so, our model demonstrates superior performance on a more complex private dataset, while they achieve better results only on high-quality datasets.
>
> **[W5]Motivation:** The theoretical foundation for using ECG as a conditioning input lies in its capacity to approximate EF to some extent. Specifically, the ratio between the first and second peaks in an ECG signal can provide a rough estimate of EF. This serves as the basis for using ECG to generate videos, and we have provide a detailed explanation of this in the Appendix. There are also works using ECG to generate reports, which also helps illustrate why we choose ECG as condition[1].
>
> However, due to inherent inaccuracies and other factors, EF estimates derived from ECG are not as precise as those from videos. This limitation means ECG cannot serve as a standalone diagnostic tool. Our video inputs inherently contain EF-related information but also provide richer temporal details, which is why we chose ECG as the conditioning input for video generation.
>
> **[W6] Table**: We have optimized the tables to improve their clarity and visual presentation. Thank you again for your constructive feedback.
>
> **Question Response:**
>
> **[Q1] Results on Pre-training only on natural dataset:** For this task, our experiments show that pretraining solely on ECHO videos leads to a significant decline in quality. This may be due to two primary reasons: first, the varying quality of ECHO videos makes it challenging to train a robust codebook; second, the limited quantity of ECHO video data restricts the codebook size, which in turn impacts performance.
>
> Another important consideration for this project is the potential privacy concerns. It is difficult to guarantee that the generated videos are entirely free of patient-identifiable information. To address this, we deliberately avoided using proprietary ECHO datasets for training the VQ-VAE. Instead, we trained it on natural video datasets, which not only helps mitigate hallucination issues but also ensures high-quality video generation.
>
> Thank you again for your suggestions; they have been incredibly helpful to us. If you have any further questions, please don't hesitate to let us know.
>
> **Reference**
>
> [1]Knight, Elizabeth, et al. "Wearable-Echo-FM: An ECG-Echo Foundation Model for Single Lead Electrocardiography." Circulation 150.Suppl_1 (2024): A4145351-A4145351.

---

### Official Review · Reviewer_31ri · 2024-10-31

**Soundness:** 3
**Presentation:** 2
**Contribution:** 2
**Rating:** 6
**Confidence:** 3

**Summary:**

The paper introduces ECHOPulse, an ECG-controlled ECHO video generation model that creates echocardiography (ECHO) videos using only ECG signals as input, claiming that it allows bypassing the need for complex conditional annotations. ECHOPulse uses a VQ-VAE-based tokenization and a transformer model to align ECG with video data, achieving efficient, realistic video generation suitable for scalable clinical applications.

**Strengths:**

- The method shows potential generalizability to other medical video modalities, like cardiac MRI and CT, indicating versatility beyond ECHO generation.
- The paper is novel and interesting
- The language and presentation is proper

**Weaknesses:**

- The use of comic sans font on the figures should re-evaluated
- High model complexity and memory demands may restrict usability on devices with limited processing power, potentially limiting real-time deployment in resource-constrained settings, such as medical ones.
- The method is over-engineered
- The premise of the paper and the approach taken actually have semantic issues. The difficulty in ECHO data comes from the poor quality of the data as the authors correctly point out, however , to the best of the reviewers knowledge, the solution is not to create synthetic to monitor patients when only ECGs are provided. Details in the videos and images have clinical significance and the proposed method makes no attempt to ensure fidelity and the trustworthiness of the generated videos, making clinical significance limited
- There is no mention of medical professionals evaluating the quality of the videos and how realistic they are
- There is no discussion about confounding factors that could influence  the generation of the videos and hence lead the model to hallucinate
- There is insufficient discussion about the use of such a video generation tool to train downstream models
- Going from an ECG to a video is an information creating process and the method makes no attempt to constrain this so the model does not generate implausible or unrelated to the patient information. Especially when there is no conditioning image/video

**Questions:**

- How would such method actually have clinical impact ?
- How much do the generated data contribute in training downstream tasks ?
- How would we ensure that the model doesn't hallucinate and provide clinically irrelevant  or harmful information ?
- Why there is no thought given to potential confounding factors that would effect the generating process ?

Overall the method is over engineered, with limited guardrails and questionable clinical significance

---

> ### Author Response · Authors · 2024-11-21
>
> We sincerely thank you for your thoughtful and constructive feedback, which has been invaluable in improving our work and clarifying its contributions. Your insights have provided important perspectives, and we are committed to addressing them to enhance the quality of our manuscript. We have submitted the revised manuscript, with all revisions highlighted in blue.
>
> **Weakness Response:**
>
> **[W1] Font issue:** Thank you very much for bringing this to our attention. We have modified all the fonts in the figures. Moving forward, we will be even more vigilant in adhering to formatting guidelines to maintain the highest level of professionalism and accuracy in our work.
>
> **[W2] Complexity Concern:** Thank you for your detailed feedback and for highlighting the need for additional parameters to complement generation speed for real-time detection. Your suggestion has helped us recognize the importance of providing more comprehensive evaluations. To address this, we will include specific details such as GPU memory usage (2.4 GB) and single-frame FLOPs (560 MFLOPs per sample). These demonstrate the feasibility of implementing our approach on wearable devices.
> Our current pipeline involves collecting ECG signals using devices such as the Apple Watch or other wearables, which are then transmitted to a smartphone for video generation[1,2]. Based on previous experience, we are confident in the practicality of this approach. Your insights will help us better communicate these aspects in the manuscript.
>
> **[W3] Over-engineered Concern:** Thank you for your thoughtful comments on model optimization. We greatly appreciate your recognition of the importance of simplifying the model structure while preserving its potential for future advancements. We chose VQ-VAE as the framework primarily because, in the rapidly evolving field of generative models, tokenizing ECHO videos can significantly benefit future research and applications. Given the potential impact, we believe that increasing the model's complexity for this purpose is a meaningful and justified decision. We will carefully consider your suggestions to optimize the overall model and reduce the number of modules.
>
> The modular design in our current approach was intended to enable future scalability, allowing components to be replaced with more advanced models as they become available. For example, we aim to train an autoregressive model to replace the existing transformer-based video generation module. Similarly, the robust VQ-VAE model was trained with the goal of enhancing the framework's potential for future improvements. Your feedback encourages us to revisit these aspects and refine our explanations in the paper.
>
> **[W4] Application Scenario:** We sincerely thank you for your insightful observations on the challenges associated with ECHO video training and your recognition of our efforts to address these issues. Your comment on the lack of high-quality annotated data is highly relevant, as it represents a key obstacle to advancing automated models in this domain. One of the major contributions of our work, as well as previous research, is to provide synthetic data to mitigate this challenge.
>
> We introduced ECG signals as conditioning inputs to enhance the realism of generated videos. This is because ECG reflects cardiac muscle motion and contains temporal information, making it more suitable than other traditional conditioning inputs (e.g., EF or segmentation masks) in terms of fidelity and trustworthiness. Introducing diffusion models to the pipeline might help. While we acknowledge that diffusion models can enhance detail by imposing additional constraints, we also recognize the trade-off in terms of increased engineering complexity and inference time. Given that our current evaluation metrics (EF estimation, FID, FVD, and expert judgment) indicate satisfactory performance, we decided not to include diffusion models in this work. However, your suggestion aligns with our long-term goals, and we are actively exploring this direction for future clinical testing. Your feedback is invaluable as we refine our strategy.
>
> **[W5] Expert Evaluation:** Thank you for your suggestion to include quantitative evaluations. We agree that providing quantitative results strengthens the credibility of our findings. To address this, we engaged clinicians from the hospital to perform quantitative evaluations, and we have include these results in the **Appendix A.4** to further substantiate our claims. Your feedback on this matter has been extremely helpful, and we are committed to ensuring our manuscript is as robust and informative as possible.
>
> **[W6] Hallucination:** Answer is the same as Q3
>
> **[W7] Downstream Task:** Answer is the same as Q2

---

> > ### Author Response · Authors · 2024-11-21
> >
> > **[W8]Video Generation Procedure:** Indeed, generating videos from ECG is inherently an information synthesis process, but it is not entirely free from confounding factors. ECG data is strongly correlated with cardiac motion, and the condition image inherently captures the complete cardiac cycle. In this context, the role of ECG is primarily to guide the transitions between different phases of the heart’s motion.
> >
> > The use of a large amount of high-quality training data ensures, to a significant extent, that our model generates accurate and meaningful information. Additionally, the observations and evaluations provided by clinicians further support the validity of our approach.
> > Furthermore, including a LoRA module in our pipeline allows us to customize outputs based on individual patient data, demonstrating the adaptability of our approach. Your encouragement reinforces our confidence in the broader applicability of our method, and we look forward to exploring this direction further.
> >
> > **Question Response:**
> >
> > **[Q1] Clinical Impact:** Thank you for raising this important concern regarding clinical relevance. Your recognition of the challenges with ECHO video quality highlights an important area that we are committed to addressing. One of the known issues with ECHO video is that its quality often depends on the operator’s expertise, and suboptimal results can arise from inexperience.  To mitigate this, our model was trained on over 90,000 high-quality ECHO videos (with plans to expand to 240,000), ensuring that the generated videos are significantly better than those captured by patients using portable ECHO devices. **We are not claiming that our model can replace any existing ECHO video acquisition methods. Instead, our goal is to propose a more portable and complementary detection approach for new scenarios, providing patients with a convenient tool for routine monitoring in daily life**. For accurate diagnosis, patients should still visit a hospital, where high-resolution equipment operated by medical professionals is essential. Additionally, our evaluation metrics confirm that the generated videos perform well on EF, the most critical metric for ECHO evaluation. This reinforces the clinical potential of our method, as emphasized in the manuscript. We asked 2 clinician to manually judge the generated videos, their comments can be seen in Appendix A.4. There is already existing work utilizing wearable devices to collect ECG data for generating reports[3]. The clinical significance of our model is essentially the same as theirs, as both aim to leverage wearable devices to obtain more information, enabling patients to conduct self-assessments. Thank you for your thoughtful comments on the importance of rigorous validation for clinical relevance. We fully agree that clinical testing is essential to establish the reliability of our method in real-world settings. To this end, we have submitted IRB documentation to the hospital and are actively working to expedite the process.  We understand that clinical validation requires substantial time and resources, but we are committed to seeing this through. For this paper, we have focused on the methodological aspects, but we aim to provide comprehensive clinical results in future work. Your encouragement reassures us that we are on the right path, and we look forward to sharing more robust findings in the future.
> >
> >
> > **[Q2]Downstream Tasks:** Thank you for your valuable feedback. Our model can be utilized to train EF estimation models, as demonstrated in the paper by comparing the results with ground truth data. This highlights the significance of our model in synthetic downstream tasks. For clinical downstream applications, such as disease detection, as mentioned earlier, additional time will be required for validation and refinement.

---

> > > ### Author Response · Authors · 2024-11-21
> > >
> > > **[Q3]Hallucination:**  We completely understand your concern. Hallucination in generative models is a well-known challenge in the field. To address this, we have taken several measures. First, we ensure the high quality of our training data. Our dataset consists of 240,000 ECHO videos, and for this study, we selected the highest-quality subset of 90,000 videos for training. High-quality training data helps reduce the likelihood of hallucination. And the VQVAE is pre-trained on 10M natural video datasets[4], which means our private ECHO dataset was not contained in VQVAE training. A VQVAE model trained on a large dataset of natural videos can significantly reduce the likelihood of hallucination. Second, detecting hallucinations remains a challenging task across the industry. At present, we rely on manual inspection. Our clinicians have conducted random evaluations of the generated videos, assessing their quality through scoring. So far, no hallucination issues have been identified. We sincerely appreciate you bringing up this point, as it is an important consideration. Moving forward, we will devote more attention to this aspect in our project and explore better approaches to detect and address hallucination. We hope to provide a more comprehensive response to this issue in the future. Thank you again for your insightful comments.
> > >
> > > **[Q4]Confounding Factors:** Thank you for your observation regarding confounding factors. We agree that this is an important consideration. In our approach, the condition image itself provides prior information about confounding factors, which allows us to account for them to some extent. Your feedback has inspired us to further investigate and articulate this aspect more clearly in the manuscript.
> > >
> > > Thank you once again for your valuable suggestions! Regarding the question on clinical applications, we can currently only provide answers based on the results of current experiments. Conducting actual clinical trials requires time. We will continue to strive and improve, and we hope to provide a better response next time.
> > >
> > > **Reference**
> > >
> > > [1] Niu, Wei, et al. "SmartMem: Layout Transformation Elimination and Adaptation for Efficient DNN Execution on Mobile." Proceedings of the 29th ACM International Conference on Architectural Support for Programming Languages and Operating Systems, Volume 3. 2024.
> > >
> > > [2]Shu, Zhihao, et al. "Real-time Core-Periphery Guided ViT with Smart Data Layout Selection on Mobile Devices." The Thirty-eighth Annual Conference on Neural Information Processing Systems.
> > >
> > > [3]Knight, Elizabeth, et al. "Wearable-Echo-FM: An ECG-Echo Foundation Model for Single Lead Electrocardiography." Circulation 150.Suppl_1 (2024): A4145351-A4145351.
> > >
> > > [4]Bain, Max, et al. "Frozen in time: A joint video and image encoder for end-to-end retrieval." Proceedings of the IEEE/CVF international conference on computer vision. 2021.

---

### Official Review · Reviewer_5Znn · 2024-11-04

**Soundness:** 3
**Presentation:** 3
**Contribution:** 3
**Rating:** 6
**Confidence:** 4

**Summary:**

This manuscript propose ECHOPULSE, a method to synthesize echo videos from easily accessible ECG signals. The method creatively uses the ECG signal, which is highly consistent with the ultrasound video, as a condition to generate the echo video. Experiments on multiple datasets and provided demos show excellent performance.

**Strengths:**

Strengths:
1) Significance: The most important contribution of this manuscript is the use of easily accessible ECG signals as control conditions to generate echo videos. The proposed method tries to combine these two modalities of video data and time series data (ECG), which is significant for medical image analysis and provides new research directions. The synthesized video data can be applied to many downstream tasks and the methods used can be generalized to more tasks.
2) Originality: Although this paper uses a several of the published techniques, the work centers on synthesizing echo videos using more temporally informative time-series data, rather than using text or otherwise.
3) Quality: The manuscript is clearly structured and accurately presented, and the figures and tables are of high quality.
4) Reproducibility: the authors provide detailed experimental procedures and details in the main text and appendices with corresponding code, which greatly helps in the reproduction of this work.

**Weaknesses:**

Weaknesses:
1) Although ECGs can be easily obtained from wearable devices, how should condition images be obtained when the method is applied in practice? How to ensure that the condition image is reasonable?
2) Compared with the previous method, the generation of real-time has been greatly improved, but still can not reach the real-time generation requirements. It is not clear how much inference resources are consumed by this method (Flops, GPU Mem et al). This determines whether it can be deployed on wearable devices.

**Questions:**

Questions:
See Weaknesses

---

> ### Author Response · Authors · 2024-11-20
>
> We sincerely thank you for your thoughtful and constructive feedback, which has been invaluable in improving our work and clarifying its contributions. Your insights have provided important perspectives, and we are committed to addressing them to enhance the quality of our manuscript. We have submitted the revised manuscript, with all revisions highlighted in blue.
>
> **Weakness and QuestionGre Response:**
>
> **[W1] Practical Applications:** Thank you for your thoughtful feedback. Your concern is something we have also considered throughout the project development phase. First, our model is capable of generating results even without a condition image. Second, patients requiring ECHO cardiac assessments typically undergo relevant tests in hospitals. Therefore, we select high-quality images with favorable angles from hospital-acquired ECHO videos as the template condition image. If patients’ data is missing, these template image may play a temporary role. Additionally, we have incorporated a LoRA module in the video tokenization stage to enable personalized customization based on a patient’s specific data. We acknowledge that the current approach is not yet perfect for clinical use, but we are actively working on improvements. Clinical applications require extensive experimentation and a lengthy validation process, and we are committed to continuously refining and enhancing our methods.
>
> **[W2] Generation Time:** Thank you for your comments. We recognize that simply improving generation speed is insufficient for real-time detection; providing additional parameters is also crucial. To address this, we provide the following details: **GPU memory usage is 2.4 GB, and single-frame FLOPs (per sample) is 560 MFLOPs.** This makes implementation on wearable devices feasible. Our current development pipeline involves collecting ECG signals using devices such as Apple Watch or other wearables and transmitting the data to a smartphone for video generation. Furthermore, we are employing model compression techniques[1,2] to further optimize the process. Based on our previous experience, this approach is viable and has the potential to achieve real-time functionality.
>
> **Reference:**
>
> [1] Niu, Wei, et al. "SmartMem: Layout Transformation Elimination and Adaptation for Efficient DNN Execution on Mobile." Proceedings of the 29th ACM International Conference on Architectural Support for Programming Languages and Operating Systems, Volume 3. 2024.
>
> [2]Shu, Zhihao, et al. "Real-time Core-Periphery Guided ViT with Smart Data Layout Selection on Mobile Devices." The Thirty-eighth Annual Conference on Neural Information Processing Systems.

---

> > ### Comment · Reviewer_5Znn · 2024-11-26
> > **weakly acceptance**
> >
> > After readin the rebuttal from the author, I retain the original score.

---

### Official Review · Reviewer_A3Vb · 2024-11-04

**Soundness:** 3
**Presentation:** 4
**Contribution:** 3
**Rating:** 8
**Confidence:** 4

**Summary:**

The authors propose the first Echocardiography generation model conditioned in ECG signals named ECHOPulse. The overall pipeline ECHOPulse consists of three stages. The first stage is video tokenizer training, which utilizes a VQ-VAE based model to train the tokenizer for ECHO video quantization. The second stage is to align the ECG and video tokens, and the ECG-FM is used to encode the ECG and masked token prediction is used for alignment. The final stage is video generation, which is autoregressive video token generation conditioned on ECG tokens.

**Strengths:**

1. The paper contains novel contribution in that it is the first paper to present Echo video generation conditioned on ECG signals, and also plan to release the ECHO videos paired with ECG dataset.

2. The paper is well-written and easy to follow, with the pipeline figure covering all three steps used in EchoPulse.

3. The experiments compare the tokenization performance and video generation performance, and compares to other related papers despite this paper being the first to explore ECG-guided ECHO video generation.

**Weaknesses:**

1. The video tokenization performance results in Table 1 compare the reconstruction metrics of MSE and MAE between EchoNet-Synthetic and ECHOPulse. However, there should also be qualitative result comparison to compare the reconstruction results between the two tokenization models, similar to qualitative video generation results in Figure 3.

2. Similar to the video tokenization weakness above, the video generation experiments also do not contain any qualitative comparison between other models such as MoonShot, VideoComposer, and HeartBeat. The figures only display the qualitative results of ECHOPulse under different ECG conditions. Therefore, qualitative results should be displayed for each models.

3. As mentioned in the Limitation and Future Works section, instead of using GAN loss for ECHOPulse, it will be better to see combination of diffusion models in this pipeline for ECHO video generation.

**Questions:**

1. As mentioned in the weaknesses section, was there a specific reason the qualitative comparison for tokenization models (reconstructed video comparison) and video generation models (generated video comparison) were not conducted?
2. Instead of using non-overlapping patches for video tokenization, would the VQ-VAE based tokenization improve performance using overlapping patches? Was this explored experimentally?

---

> ### Author Response · Authors · 2024-11-20
>
> We sincerely thank you for your thoughtful and constructive feedback, which has been invaluable in improving our work and clarifying its contributions. Your insights have provided important perspectives, and we are committed to addressing them to enhance the quality of our manuscript. We have submitted the revised manuscript, with all revisions highlighted in blue.
>
> **Weakness Response:**
>
> **[W1] Qualitative Comparison:** Thank you for your valuable suggestions. According to your suggestion, we have added the comparison of reconstruction results with EchoNet-Synthetic in Table 1. And we have added Figure 11 in Appendix to show the reader the visualization results of model’s reconstruction performance.
>
> **[W2]Lack of Video Generation Comparison:** Thank you for your insightful feedback. We have added additional figures in the appendix, which compares our method with current models to further emphasize our model’s privilege.
>
>
> **[W3] Add Diffusion Model to Pipeline:** We appreciate your suggestion. Other reviewers have expressed similar concerns. We have indeed considered incorporating diffusion models into the pipeline to impose stronger constraints on video generation and improve video quality. However, we are somewhat concerned that this may overly complicate the video generation process. Moreover, the current results, both in terms of quantitative metrics and expert evaluations, do not reveal significant issues. Nevertheless, we are actively exploring and researching this direction.
>
> **Questions Response:**
>
> **[Q1]Qualitative Comparison:** Thank you for your comments. While our tokenization model was designed and trained with significant effort, it is not the primary contribution we aim to highlight in this paper. The comparisons with other ECHO video generation methods are intended to demonstrate that our video tokenization model is effective and performs on par with established approaches. The success of this paper does not primarily rely on the performance of the video tokenization model. Other video generation models based on VQ-VAE, such as Phenaki[1], MAGVIT[2], and MAGVIT2[3], are not open-sourced, making it impossible for us to compare with them.
>
> **[Q2]Exploration of Over-lapping Patches:** Regarding the suggestion about overlapping patches, we have indeed experimented with similar approaches. However, for this task, the performance improvement was not significant, and we did not observe artifacts in the current results. Additionally, given that the VQ-VAE codebook requires searching, overlapping patches would increase the computational burden. For these reasons, we decided not to further pursue this direction at the time.
>
> Thank you again for your suggestions; they have been incredibly helpful to us. If you have any further questions, please don't hesitate to let us know.
>
> **Reference:**
> [1] Villegas, Ruben, et al. "Phenaki: Variable length video generation from open domain textual descriptions." International Conference on Learning Representations. 2022.
> [2] Yu, Lijun, et al. "Magvit: Masked generative video transformer." Proceedings of the IEEE/CVF Conference on Computer Vision and Pattern Recognition. 2023.
> [3] Yu, Lijun, et al. "Language Model Beats Diffusion--Tokenizer is Key to Visual Generation." arXiv preprint arXiv:2310.05737 (2023).

---

### Meta-Review · Area_Chair_mSkZ · 2024-12-21

**Metareview:**

This paper introduces ECHOPulse, an ECG-conditioned echocardiogram video generation model leveraging VQ-VAE tokenization for efficient synthesis, enabling controllable, high-quality medical video generation and advancing multimodal cardiac monitoring applications.

Strengths
- Reviewers agree that the paper introduces a novel approach by conditioning ECHO video generation on ECG signals.
- Echopulse demonstrates state-of-the-art performance in both quantitative and qualitative measures for video generation and shows potential generalizability to other medical imaging modalities.
- Echopulse is implemented in an efficient way to support real-time applications, including resource usage metrics and potential deployment on wearable devices.

Weaknesses
- Reviewers are not entirely convinced of the clinical significance and reliability of the generated videos.
- Some of the metrics (FID, SSIM) used for evaluation didn’t seem suitable, and baseline models could be stronger.
- Some of the descriptions regarding the limitations of previous work are inaccurate and oversimplified.

**Additional Comments On Reviewer Discussion:**

Clinical relevance and potential hallucination were questioned, to which the authors responded with manual inspection by clinicians. Reviewers were also concerned with some baseline models missing such as EchoNet and EchoDiffusion, which authors added in the experiments.

---

### Decision · Program_Chairs · 2025-01-22

Accept (Poster)